

**Sonar Gas Flux Estimation by Bubble Insonification: Application to Methane Bubble Fluxes**
**from the East Siberian Arctic Shelf Seabed**
Ira Leifer[1,], Denis Chernykh[2,3], Natalia Shakhova[3,4], Igor Semiletov[2,3,4]
[1] Bubbleology Research International, Solvang, CA 93463
[2] Russian Academy of Science, Pacific Oceanological Institute, Vladivostok, Russia
[3] National Tomsk Research Polytechnic University, Tomsk, Russia
[4] University Alaska Fairbanks, International Arctic Research Center, Fairbanks, USA
**Abstract**
Sonar surveys provide an effective mechanism for mapping seabed methane flux emissions, with Arctic
submerged permafrost seepage having great potential to significantly affect climate. We created *in situ*
engineered bubble plumes from 40-m depth with fluxes spanning 0.019 to 1.1 L/s to derive the *in situ*
calibration curve, $Q(\sigma)$. Non-linear curves relating flux, $Q$, to sonar return, s, for a multibeam
echosounder (MBES) and a single beam echosounder (SBES) for a range of depths demonstrated
significant bubble-bubble acoustic interactions – precluding the use of a theoretical calibration function,
$Q(\sigma)$, wherein bubble $\sigma(r)$ scales with the radius, $r$, size distribution. Bubble plume sonar occurrence,
$\Psi(\sigma)$, with respect to $Q$ found $\Psi(\sigma)$ for weak $\sigma$ well described by a power law that likely correlated with
small bubble dispersion and strongly depth dependent. $\Psi(\sigma)$ for strong s largely was depth-independent,
consistent with bubble plume behavior where large bubbles in a plume remain in a focused core. As a
result, $\Psi(\sigma)$ was bimodal for all but the weakest plumes.
$\Psi(\sigma)$ was applied to sonar observations of natural arctic Laptev Sea, seepage including accounting for
volumetric change with a numerical bubble plume. Based on MBES data, values of total $Q_m$, the mass
flux, were 5.56, 42.73, and 4.88 mmol/s with good to reasonable agreement between the SBES and
MBES data (4-37%) for total $Q$. Seepage occurrence, $\Psi(Q)$, was bimodal, with weak $\Psi(Q)$ in each seep
area well described by a power law, suggesting primarily minor bubble plumes. Seepage mapped spatial
patterns suggested subsurface geologic control attributing methane fluxes to the current state of subsea
permafrost.
**Keywords:** Bubble, multibeam sonar, single beam, quantification, Arctic, methane, submerged
permafrost, field study, seep, engineered bubble plume



**1. Introduction**
**1.1 Arctic methane and climate change**
*Methane and Arctic climate change*
The second-most important anthropogenic greenhouse gas on a century timescale after carbon dioxide,
$CO_2$ is methane, $CH_4$, (Forster et al. 2007); however, on decadal time scales comparable to its
atmospheric lifetime, $CH_4$ is more important to the atmospheric radiative balance than $CO_2$ (IPCC, 2007;
Fig 2.21). After nearly stabilizing, atmospheric concentrations began increasing a decade ago, although
the underlying reasons remain poorly understood (Nisbet et al., 2014). Despite likely increasing future
natural emissions from global warming feedbacks (Rigby et al., 2008) and anthropogenic activities
(Kirschke et al., 2013; Wunch et al., 2009), many current source estimates have large uncertainties with
greater uncertainty in future trends, particularly in the Arctic.
Currently, Arctic global warming is the strongest, termed Arctic amplification (Graversen et al., 2008).
Permafrost $CH_4$ provides an important feedback, where warming Arctic temperatures release $CH_4$
sequestered in and under terrestrial (Friedlingstein et al., 2006; Lemke et al., 2007) and sub-sea
permafrost, which is submerged terrestrial permafrost (Shakhova and Semiletov, 2009). Sediment
accumulation rates for the Arctic continental shelf are 5 times greater than elsewhere in the World's
Oceans. For example, sedimentation for the Siberian Arctic shelf where the six Great Siberian Rivers
outflow, has deposited organic carbon into bottom sediments that approximately equals accumulations
over the entire pelagic area of the World's Oceans. This leads to the thickest (up to 20 km) and most
extensive sedimentary basin in the world, the "Arctic super carbon pool" (Gramberg et al., 1983).
*The Siberian Arctic Shelf*
The Siberian Arctic Shelf subsea permafrost, $CH_4$ hydrates, and natural gas systems contains vast $CH_4$
deposits (Gautier et al., 2009; Gramberg et al., 1983; Romanovskii et al., 2005; Serreze et al., 2009;
Shakhova et al., 2009a, 2010a; 2010b) of which a large fraction is $CH_4$ hydrate deposits (Makogon et al.,
2007; Soloviev et al., 1987). According to Dickens (2003), subsea continental shelf reservoirs are
estimated to contain about 10,000 gigatonnes Gt (1 Gt=$10^{15}$g) of $CH_4$ hydrates, compared to terrestrial
permafrost, which is estimated at 400 Gt of $CH_4$ hydrates. The Arctic continental shelf makes up 25% of
the entire area of the world's oceanic continental shelves (7 million $km^2$ of the ocean's area, 28.8 million
$km^2$) and is estimated to contain 2,500 Gt of carbon in the form of $CH_4$ hydrates. This is more than 3
times greater than the amount of carbon currently stored in the atmosphere and ~500 times greater than
the current atmospheric $CH_4$ reservoir (IPCC, 2007). Remobilization of only a small fraction of $CH_4$ in
these deposits could trigger abrupt climate warming; for example, Archer and Buffett (2005) estimated





that release to the atmosphere of just 0.5% of the $CH_4$ in Arctic shelf hydrates could cause abrupt climate
change.
The East Siberian Arctic Shelf (ESAS) is the world's largest and shallowest shelf (covering $2.1 \times 10^6$ km$^2$)
containing the largest area of submerged permafrost by far (Shakhova et al., 2010a, b). The ESAS is a
seaward extension of the Siberian tundra that was flooded during the Holocene transgression, 7-15 kyr
ago (Romanovskii et al., 2005). The ESAS comprises ~25% of the Arctic continental shelf and contains
over 80% of existing subsea permafrost and shallow hydrate deposits, estimated at ~1400 Gt carbon
(Shakhova et al., 2010a). This reservoir includes ESAS hydrate deposits, estimated at ∼540 Gt of $CH_4$,
with an additional 2/3 (∼360 Gt) trapped below as free gas (Gramberg et al., 1983; Soloviev et al., 1987).
ESAS subsea permafrost is Siberian terrestrial permafrost that was submerged and thus is expected to
contain similar permafrost organic carbon (OC) deposits to terrestrial, implying a further 500 OC Gt
within an ~25-m thick permafrost layer. Thus, ESAS carbon stores are comparable to the Arctic soil
carbon pool, which includes tundra and taiga (~1000 Gt C) and coastal permafrost (~400 Gt C) (Tarnocai
et al., 2009).
The ESAS subsea permafrost is changing in response to glacial/interglacial Arctic warming (~7°C), and
warming from the overlying seawater (~10°C) since inundation in the early Holocene, with additional
ESAS seawater warming in recent decades (Biastoch et al., 2011; Nicolsky et al., 2012; Semiletov et al.,
2012, 2013;Shakhova et al., 2014). The Siberian rivers transport additional heat to the Arctic shelf which
results from terrestrial ecosystem responses to global warming. This includes the degradation of terrestrial
permafrost and increased river runoff, which warms shelf waters. In turn, this warm runoff drives a
downward heat flux to shelf sediments and sub-sea permafrost (Shakhova and Semiletov, 2007; Shakhova
et al., 2014). Also, there is the potential for abrupt $CH_4$ release on the ESAS and its continental slope
related to temperature destabilization of Arctic shallow and oceanic hydrates. The extent of the ESAS gas
hydrate stability zone is expected to be highly sensitive to small temperature changes (Dickens, 2003).
*Permafrost Degradation*
Subsea permafrost is an impermeable lid (where continuous) preventing the upward migration of $CH_4$ and
other geological fluids, hence the great concern for its degradation and sub sequent release of sequestered
$CH_4$ to the shallow ocean and then atmosphere. Both onshore and offshore Arctic permafrost degrade
from two directions (Ostercamp, 2001; Shakhova and Semiletov, 2009). Thawing occurs from the top
downward, where the active layer expands downward creating taliks (bodies of thawed permafrost).
Permafrost also degrades from the bottom up as a result of geothermal heat flux, where heat from the
Earth's interior flows upward, thawing frozen sediments from below. The latter only has a significant



effect for submerged offshore permafrost (Romanovskii et al., 2005), because the high Arctic terrestrial
permafrost is thick and continuous, allowing its bottom to absorb upward heat flows with negligible
effect. For example, an offshore permafrost sediment core (obtained by authors' team from the fast ice in
April 2011 to 57 m below the Laptev sea floor) was unfrozen and 8-12°C warmer than a core recovered
from the Lena Delta' borehole (Shakhova et al., 2014).
Recent studies have identified four main subsea permafrost degradation mechanisms, which provide
geologic control of the thermal state of subsea permafrost and also hydrate stability. The most important,
which operates on long (millennia) timescales, is the increasing temperature of the overlying bottom
seawater and the duration of its interaction with the permafrost both by heat transfer and salinization
(Soloviev et al., 1987; Shakhova et al., 2014, 2015). A second process that provides geologic control
arises from heating from large Siberian rivers which drives bottom water warming and is proposed to
control the distribution of open taliks in coastal ESAS waters (Shakhova et al., 2014). Thirdly, high
geothermal heat flow in rift zones induces fractures that provide geologic control (Drachev et al., 2003;
Nicolsky et al., 2012). Finally, areas of high heat flow includes relic–thaw lakes and river-valleys that
were submerged during the Holocene inundation, but still drive modern permafrost degradation (Nicolsky
and Shaklhova, 2010; Nicolsky et al., 2012; Shakhova et al., 2009b, c).
Subsea permafrost degradation is greatest in the outer shelf waters, which are deeper than 50 m, where
submergence at the beginning of Holocene (~10-15 thousands years ago) first occurred (Bauch et al.,
2001) and where4 current models predict discontinuous and mostly degraded permafrost in the outer
Laptev Sea. The formation and growth of subsea thaw lakes also likely is greater where riverine heat
inputs combines synergistically with longer permafrost submergence (Shakhova and Semiletov, 2007;
Holemann et al., 2011; Shakhova et al., 2014). This also leads to the evolution and growth of taliks, which
provide effective gas migration pathways to the shallow waters of the ESAS (Nicolsky and Shakhova,
2010; Shakhova et al., 2009b, 2014, 2015; Nicolsky et al., 2012). River outflow also affects ocean
temperatures by introducing colored dissolved organic matter (CDOM), which concentrates absorption of
solar radiation in near surface waters, accelerating ocean warming, freshening, and acidification (Pugach
et al., 2015; Semiletov et al., 2013, 2016).
Geologic heat flow is strong in the Laptev Sea (85-117 m W m$^{-2}$) where active seafloor spreading is
converting into continental rifting. In fact, the northern Laptev Sea also is one of the few places where
active oceanic spreading approaches a continental margin (Drachev et al., 2003) and correlates with the
"hot" area crossed by the Ust' Lensky Rift and Khatanga-Lomonosov Fracture (Drachev et al., 2003;
Nicolsky et al., 2012). Evidence for this rifting is provided by hydrothermal fauna remnants documented



around grabens (dropped blocks between faults) in the up-slope area that typically occur along oceanic
divergent axes (Drachev et al., 2003). Grabens in the ESAS often manifest as linear structures and also
often correlate spatially with paleo-river valleys.
Migration from this submerged permafrost reservoir to the seabed feeds a vast marine seep field entirely
in shallow waters, where emissions contribute directly to the atmospheric budget (Shakhova et al., 2014).
At-sea observations show dissolved $CH_4$ supersaturation with respect to the atmosphere for >80% of
ESAS bottom waters and >50% of surface waters (Shakhova et al., 2010a, 2010b). This seepage is almost
entirely ancient $CH_4$ – modern $CH_4$ production from old OC is negligible based on recent microbiological
studies (2011-2012) in ESAS surface and long-sediment cores (V. Samarkin, unpublished data). Indeed,
in the ESAS, sediment OC content varies by a factor of ~4, while ebullition $CH_4$ fluxes vary by orders of
magnitude (Shakhova et al., 2015).

### 1.2 Study motivation

Given the extent of Arctic seepage and the magnitude of current and potential future emissions, there is a
critical need for new approaches to effectively, rapidly, and quantitatively survey large seepage areas.
Video is inadequate to survey extensive or widely dispersed seepage, a task for which sonar (active
acoustics) excels. This study's motivation is to demonstrate an improved approach for seabed seepage
survey in the Arctic, using *in situ calibrated* sonar-derived bubble fluxes.
Herein, we present *in situ* calibration experiments in the ESAS to investigate the evolution of bubble
plume sonar return (multiple beam echosounder–MBES and single beam echosounder–SBES) from rising
engineered bubble plumes spanning a broad range of flow rates covering typical seepage bubble flows to
infer the relative importance of small and large bubbles to sonar return signatures. Analysis demonstrated
that bubble-bubble acoustic interactions are *non-negligible* for the first 15 m of rise at least, preventing
simple flux inversion by dividing total sonar return by the sum of individual bubble sonar cross sections
for an assumed bubble size or size distribution.
The calibration curves then were applied to quantify *in situ* sonar observations of three areas of active
natural bubble seepage nearby the site of the calibration experiments. Because the calibration and seep
bubble plumes were different gases and from different water depths, with slightly different temperature
profiles, bubble dissolution rates are different – i.e., for the same seabed mean volume flux, the depth-
window-averaged volume fluxes are different. We make a first attempt to correct for this factor by
applying a numerical bubble-plume model initialized with a typical seep bubble plume size distribution to
the two bubble flows (calibration and natural seepage).





### 1.3. Marine seepage

Marine seepage is a global phenomena where $CH_4$ and other trace components escape as bubbles from the
seabed and rise towards the sea surface (Judd and Hovland, 2009), dissolving and depositing $CH_4$ in the
water column while transporting their remaining contents to the sea surface – if they do not dissolve
subsurface (Leifer and Patro, 2002b).
In the shallow waters, like the Coal Oil Point (COP) seep field, most of the $CH_4$ reaches the atmosphere
directly (Clark et al., 2005) from mixing in the near field (Clark et al., 2000) and in the far (down-current)
field when winds strengthen as typical occurs diurnally in coastal California. The ultimate fate of
dissolved seep $CH_4$ depends most strongly on its deposition depth (Solomon et al., 2009) with $CH_4$ below
the Winter Wave Mixed Layer (WWML) largely being oxidized microbially (Rehder et al., 1999).
However, even for deepsea seepage (to ~1 km), field studies show seep bubble-plume transport of $CH_4$ to
the upper water-column and atmosphere (MacDonald, 2011) due to plume processes (Leifer et al., 2009)
and hydrate skin phenomena (Rehder et al., 2009; Warzinski et al., 2014). Still for deepsea seepage, a
significant fraction of seabed $CH_4$ emissions are deposited below the WWML where they are oxidized
microbially. In the shallow ESAS, virtually all the seabed $CH_4$ (dissolved and gaseous) is emitted in the
WWML and escapes to the atmosphere (Shakhova et al., 2014). However, even $CH_4$ dissolved below the
Arctic WWML is less likely to be oxidized than in non-Arctic waters because Arctic $CH_4$ oxidation rates
are very slow, 300-1000 days (Shakhova et al., 2015). These slower rates allow release of some of this
deeper aqueous inventory to the atmosphere during storms and fall-winter convection (Shakhova et al.,
2010a, 2014).
Plume processes are important. Several factors control the fate of seep bubble $CH_4$ including depth,
bubble size, plume volume flux (Leifer et al., 2006; Leifer and Patro, 2002b), and the vertical bubble
velocity, $V_{up}$, which includes the upwelling flow and depends on the plume volume flux (Leifer, 2010;
Leifer et al., 2009). Another documented important plume process is enhanced aqueous concentrations
relative to the surrounding water, which enhances bubble survival (Leifer et al., 2006).

### 1.3 Seep Bubble Measurements

Currently, natural seepage bubble-plume size distributions, $\Phi$, have only been measured by video and
passive acoustics (Leifer, 2010), with the latter demonstrated only for low-emission-rate bubble plumes
where the individual bubbles acoustic signatures can be identified (Leifer and Tang, 2007). Although
highly accurate, video requires significant power, data storage, and its analysis is complex; nor is it a
remote sensing technique (Leifer, 2010)–i.e., currents shift bubbles out of the measurement volume.





Most natural seepage bubbles fall within a relatively narrow size range. Specifically, based on a review of
39 bubble-plume size distributions (the most comprehensive to date), Leifer (2010) found that the vast
majority of seep bubble plumes could be classified in two primary categories, termed major and minor,
with the latter most common, a characterization found in other studies, reviewed in Leifer (2010).
$\Phi$ for minor bubble plumes are well described by a Gaussian function and comprised of bubbles largely in
a narrow size range, $1000 < r_e < 4000$ µm, where $r_e$ is the equivalent spherical radius. Major bubble
plumes generally escape from higher flow vents as a fragmenting gas jet with a power law size
distribution. Most major bubble plumes are small; however most of the plume volume is transported by
the largest bubbles, up to $r \sim 1$ cm.
Video bubble measurement is highly local and thus a poor survey (or monitoring) tool. A hybrid ROV
video approach was demonstrated by Leifer (2015) for an 1100 m² North Sea seep site where the ROV
was flown in a grid pattern. A total of 176 bubble plumes were classified by appearance and assigned an
emission flow, which was integrated for the entire site (estimated at 440 plumes), with the strongest
plume class's flux measured directly in the field. The video survey required about a full day of ROV dive
time and analysis was labor intensive.

**1.4 Sonar Seep Bubble Measurements**

Reported seepage areas span a large range of spatial areas and number of plumes. Sonar has been used to
survey concentrated seep area covering ~1000 m² in the North Sea noted above (Schneider von Deimling
et al., 2010; Wilson et al., 2015), and far more dispersed and weaker seepage in the Black Sea of ~2500
plume in an areas of ~20 km² (Greinert et al., 2010). Significantly larger and stronger seepage in the COP
seep field, offshore California have been mapped by sonar too. The COP seep field covers ~3 km² of
active seabed in an 18 km² area (Hornafius et al., 1999), and comprises tens of thousands of plumes. The
COP seep field includes highly focused seepage, termed megaseeps, which release more than a million
liters per day (Washburn et al., 2005). Megaseeps may arise from dozens to thousands of vents. Seepage
on far larger scale exists in the ESAS where ~30,000 plumes were identified manually in just two
transects. Seepage densities as high as ~3000 seep bubble plumes per km² were found transecting a single
hotspot. Based on the hotspot size (18,400 km²), an order of magnitude estimate suggests 60 million seep
plumes in the hotspot alone. While sonar surveys of a localized site, e.g., the North Sea site, can be
conducted in a few minutes, the ESAS sonar survey required a month for two transects (Stubbs, 2010;
Shakhova et al., 2014).
Sonar is highly effective at seep emission mapping; however interpretation challenges exist even for
qualitatively assessment of relative emission strength. For SBES systems, there is geometric uncertainty –





the plume's angular location is unknown; a problem resolved by MBES systems (Leifer et al., 2010). In
addition, sonar (SBES or MBES) loses fidelity from multiple plumes in close proximity (Schneider von
Deimling et al., 2011; Wilson et al., 2015) where the sonar returns along multiple pathways, creating
ghosts, shadow noise, off-beam returns, scattering loss, and other artifacts (Wilson et al., 2015). Note, if
bubble spatial densities are sufficiently high for artifacts to occur between plumes, then they are
sufficiently high to produce artifacts within plumes between individual bubbles. For very high flux bubble
plumes, the sonar return signal can be largely or even completely lost (Leifer et al., 2010). In addition, the
vessel's acoustic environment can be challenging both acoustically and from electrical noise.
Furthermore, the ocean is far from acoustically transparent, with signal loss and scattering from
suspended sediment and biota, often in layers, as well as other marine acoustic features.
Although seemingly straightforward, there are many challenges to quantitative derivation of bubble
emission flux from sonar return, which at its basis relates to the interaction of sound with a bubble. For a
single spherical bubble the relationship has long been known, with resonance given by the Minnaert
(1933) equation:
$$f_o = \frac{1}{2\pi r}\left(\frac{3\gamma P}{\rho}\right)^{1/2} \tag{1}$$
where $f_o$ is the resonance (or Minnaert) frequency, $\gamma$ is the resonance (or Minnaert) frequency, $P$ is
pressure, and $\rho$ is pressure, and for Minna non-spherical bubbles ($r > 150$ µm) an eccentricity correction
is needed to account for the angle between the bubble axes and the sound wave front. Bubble
eccentricities vary from 1.0 for spherical bubbles to 2 or greater for $r > 3500$ µm (Clift et al., 1978). For a
single spherical bubble, the back-scattering cross section, $\sigma_B$, near $f_o$ is (Weber et al., 2014):
$$\sigma_B = \frac{r^2}{\left[\left(\frac{f_o}{f}\right)^2 - 1\right]^2 + \delta^2} \tag{2}$$
where $f$ is frequency and $\delta$ is the damping term that can be approximated as $\delta \sim 0.03 f^{0.3}$ with $f$ in kHz.
From, here, integrating over the bubble emission size distribution, $\Phi(r)$, which is the number of bubbles
in a radius bin, $r$, passing through the measurement plane, combined with $V_Z(r)$, the bubble vertical
velocity, which is a function of $r$, over the measurement volume yields the total plume cross-section if
bubbles are acoustically non-interactive and scattering is isotropic.

.



Scattering is radially symmetric about the plume axis, θ, but not in the azimuth, β. Thus, for
ellipsoidal bubbles in a bubble plume observed from angle β, the scattered power, $P_{Bs}$, is
$$P_{bs}(\beta) = \iiint_{x,y,r} \sigma_B(\beta)\Phi(x,y,z,r_e) + \iiint_{x,y,r,k,m} G_{k.m}\sigma_{k.m} \qquad (3)$$
where $k$ and $m$ are indices of different bubbles ($k \neq m$) and $\Phi(r,z)$ changes with altitude above the seabed
due to dissolution and air uptake, and can vary horizontally with position in the plume due to currents and
the complex fluid motions associated with bubble plumes (Asaeda and Imberger, 1993; Leifer et al.,
2009). The equation includes a second acoustic interaction term and is the integral over bubble$_k$-bubble$_m$
interactions (multiple scattering and acoustic coupling). This term is described by the interaction function,
$G$, and depends on the bubble sizes, separation distance and angle, etc. In the case of a sufficiently
dispersed bubble plume (large bubble-bubble separation), $G_{k,m}=0$ and bubbles are acoustically non-
interacting.
In most seep bubble plumes, the close proximity between bubbles creates bubble-bubble acoustic
interactions through acoustic coupling and/or multiple scattering. Multiple scattering occurs when the
sound scattered from one bubble interacts and scatters from a second bubble back in the direction of the
sonar receiver. The significance of multiple scattering is provided by artifacts like ghosting between
plumes (not return from the sonar beam sidelobes).
Acoustic coupling occurs for bubbles within 10-20 bubble radii of each other, i.e., a few centimeters, such
that the water surrounding the bubble no longer is incompressible, leading to a frequency shift (Leifer and
Tang, 2007). Because sonar is spectrally selective, frequency shifts from acoustic coupling can decrease
the sonar return signal. In most seep bubble plumes, acoustic coupling should be small except very near
the seabed where bubbles still rise in close proximity, or where bubbles rise in dense clumps. In the latter
case, smaller bubbles often draft larger bubbles and remain in close proximity (Tsuchiya et al., 1996).
**2. Methodology**
**2.1. Coal Oil Point seep field Scoping Study**
A precursor study was conducted in the COP seep field (Fig. 1) prior to the Arctic field experiment to
demonstrate 4D seep monitoring by a scanning MBES. The rotator-lander was deployed ~15 m from the
center of Shane Seep, which covers an area of ~$10^4$ m$^2$ in ~20-m water depth and comprises on the order
of 1000 individual vents or bubble plumes (Fig. 1B).



The lander included a MBES (DeltaT, Imagenex, Vancouver, Canada) and compass (Ocean Server, MA)
on an underwater rotator (Sidus Solutions, CA) with azimuthal rotation of up to 270° angle range. The
sonar produced a vertically oriented 128-beam fan spanning 120°, tilted upwards to reduce seabed
backscatter. Two *in situ* calibration air bubble flows were deployed ~8 m from the lander at azimuthal
angles beyond the active seepage area and were traversed during each sonar rotation cycle. Regulated
airflow from an onboard compressor fed these bubble plumes and was measured by two rotameters.

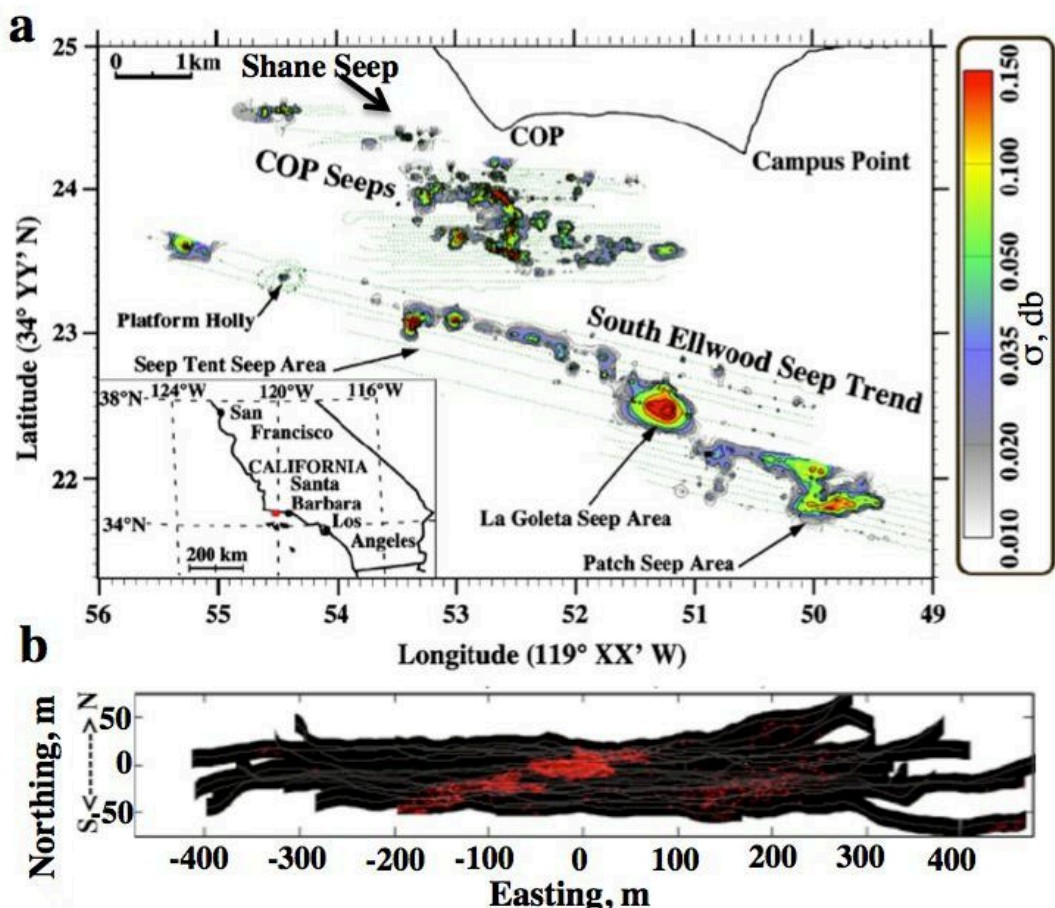


**Figure 1. a.** Coal Oil Point (COP) seep field map, showing Shane Seep location where the scoping study
was performed based on 2005 sonar data. From Leifer et al. (2010). **b.** Multibeam sonar survey map (2-m
depth window at a seabed-following height of 4 m) of Shane Seep in the COP seep field, collected in

2009.



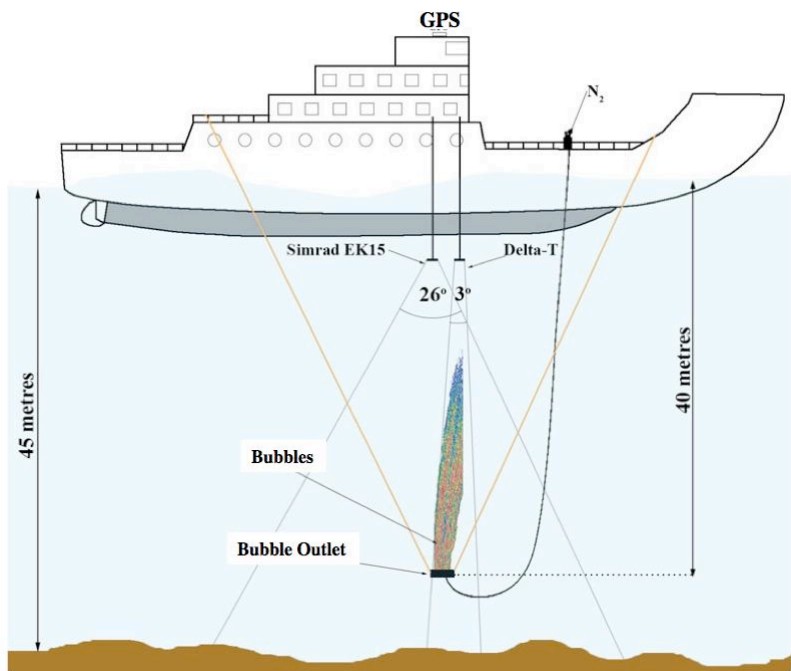


**Figure 2.** *In situ* calibration experiment set up schematic as deployed in the ESAS.
**2.2. Arctic Calibration Experimental Set-up**
For the control sonar bubble plume study, bubble plumes were made from nitrogen supplied by a pressure
tank on the vessel foredeck. A 70-m long, 12-mm diameter, 6-mm wall thickness, air supply tubing was
attached by a Kevlar rope to a heavy metal weight (~30 kg) that ballasted against buoyancy of air in the
tubing and drag from currents. The supply tube was deployed to 40-m depth in water of ~45-m depth
(Fig. 5) and the rising bubble plume was observed with MBES and SBES. The sonars were located near
each other so that their beam coverage overlapped with the center beam focused on the end of the bubble
stream. Bubbles were produced from a 4-mm diameter copper nozzle attached at the end of the air supply
tube.
Gas flow was controlled using standard flow meters, one port of which was connected to a PVC tube and
another was connected to a 2-way valve, the second port of which was connected to the gas tank through
the gas manifold. The manifold consisted of a high-pressure sensor of the tank pressure and a low-
pressure sensor for the out-coming pressure (5.5 bar). We used temperature-compensated differential
pressure sensors with a manufacturer-specified range of ±1 psi (equivalent to ±70 cm of water). The
sensor has manufacturer-specified accuracy and stability of ±0.5% FSD (full scale deflection over the
operating pressure range of the sensor over 1 yr, between 0 and 50°C) and repeatability errors of ±0.25%




FSD. For the study, the gas flow was varied from 0.5 to 150 L min$^{-1}$ at 5.5 bar (equal to the bubble outlet
hydrostatic pressure). For each experiment, the gas flow was allowed to stabilize and then sonar data were
recorded for ~10 minutes.
SBES bubble plume data were collected by a SIMRAD EK15 SW 1.0.0 echosounder (www.simrad.com)
at 200 kHz, with a 1 ms pulse duration at 10 Hz, 26° beam width, and built-in calibration system. MBES
bubble plume data were collected by a DeltaT Profiler (Imagenex, British Columbia, Canada) at 260 kHz.
Sonar data including seep bubble plumes were recorded at an average survey speed of 4-6 knots. Sonar
backscatter was calibrated using acoustic targets (SIMRAD, Denmark). Initial data visualization and
processing used EchoView and Sonar5 software (SIMRAD), for the EK15.
**2.2 Arctic Field Campaign**
Field data were obtained during an expedition onboard the research vessel R/V *Victor Buynitsky* from 2
Sept. to 3 Oct. 2012 (Figs. 3 and 4). The R/V *Victor Buynitsky sailed* from Murmansk to the Laptev Sea
and the adjacent portion of the ESAS. The weather during the expedition was typical for this region for
this time of the year (3-4 storm events with wind speed >10 m s$^{-1}$). The calibration experiment was
performed in the Kara Sea for 45-m water depths under favorable weather: calm sea with wind speed 3-5
m s$^{-1}$ and wave height of 0.2-0.5 m. The expedition's overarching goal was to improve understanding of
the current scale of ESAS $CH_4$ emissions in order to develop a conceptual model of $CH_4$ propagation from
the seabed to the atmosphere, including assessing source strengths and their dynamics.

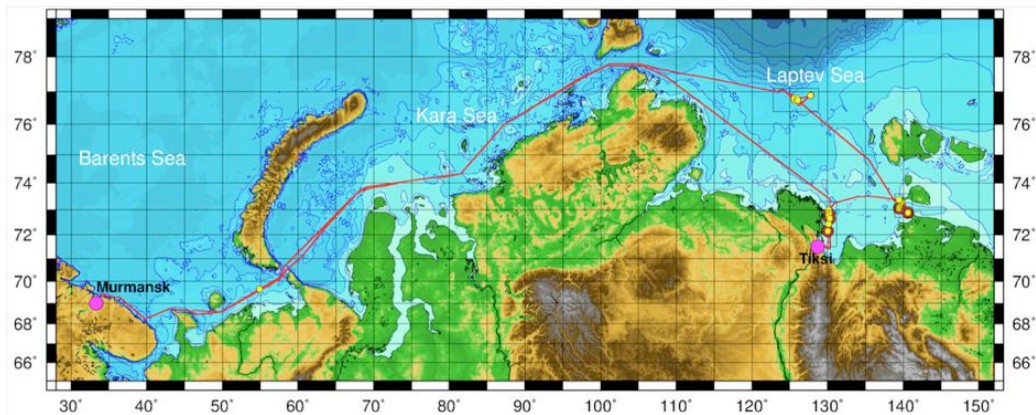


**Figure 3.** Map for R/V *Victor Buynitsky* cruise, 2012.




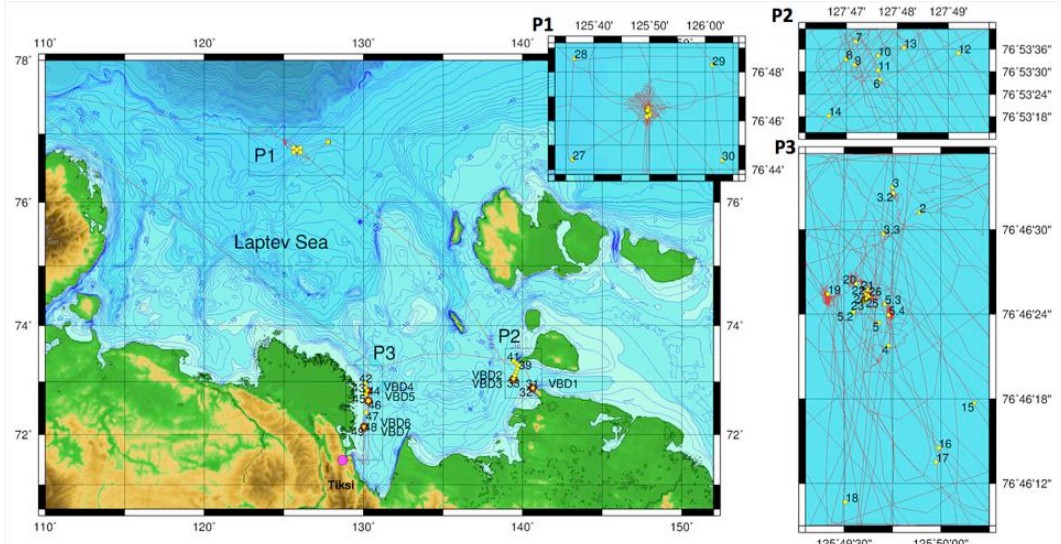


**Figure 4**. Locations of oceanographic stations for RV *Victor Buynitsky* cruise, 2012, marked by yellow
circles. Polygons of major focus areas are marked as P1 (northern Laptev Sea), P2 (east Lena Delta) and
P3 (Dmitry Laptev Strait), shown in insets. Ship tracks accompanied by CTD measurements (and
geophysical survey) performed in the P1 are shown as red lines.

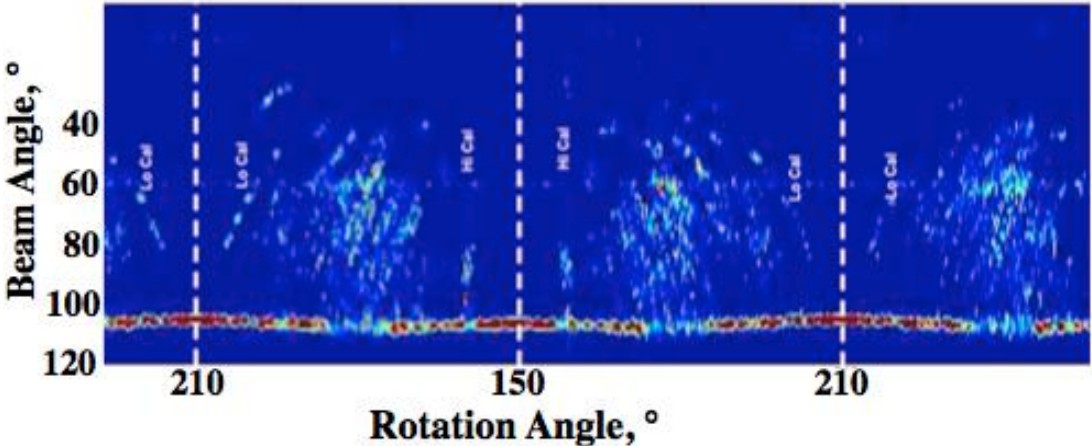

**Figure 5.** Spherical (time) slices showing Shane Seep and two calibration air flows, labeled on figure
from a sonar rotator lander deployment in 2008. Sonar was deployed ~8 m southeast from the main seep
bubble plume (see Fig. 1b, relative to origin).





## 3. Results

### 3.1. COP Seep Field Precursor Study

The importance of bubble-bubble acoustic interaction was demonstrated for calibration plumes during the scoping study sonar experiment,. Example MBES rotator data are shown in Fig. 5 for a little over a rotation cycle, which includes the main seep and both the high and low calibration bubble plumes. Arctic weather challenges limited the rotator lander deployment to just a few hours.

Sonar returns for the two calibration plumes (Fig. 5) were thresholded above background (bubble-free water) and integrated for each beam during rotation across each calibration plume. The thresholded sonar return in a depth window then was fit with a linear polynomial of the log of the integrated sonar return over the plume, σ, versus height, $h$ (Fig. 6). The value of σ increased as the bubble plume rose – i.e., σ($h$) was not constant – even though air bubble volume change is minimal over such short rise heights. This is evidence of bubble-bubble acoustic interaction decreasing as the bubbles rise and spread from turbulence (acoustic interactions decrease towards zero as the inter-bubble distances increases to large distances).

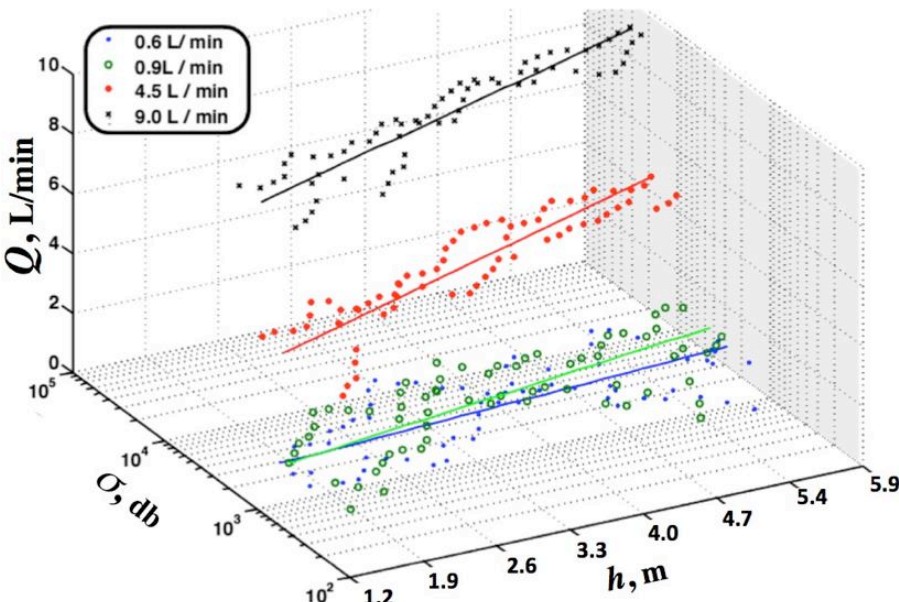

**Figure 6.** Field sonar calibration return, σ, from the Coal Oil Point seep field for air bubbles in 22-m deep water. Sonar return integrated across the plume, σ, versus airflow, $Q$, and height above seabed, $h$, for four airflows and least-squares linear-regression fits to log(σ) versus $h$.





### 3.1. Calibration


*In situ* calibration experiments were conducted in the Kara Sea (neighboring the Laptev Sea) in a region
of no natural seepage and almost flat seafloor to reduce or eliminate off-beam acoustic seabed scattering.
Winds were unusually calm for this region, 1-3 m s$^{-1}$, with no significant waves (0 to 1 ball). Column
profile temperature and salinity data were obtained by a conductivity temperature depth, CTD (SBE19+,
Seabird, USA). The vessel was anchored during the calibration experiments. The wave-mixed layer
(WML) extended to ~35 m depth, with upper water warmer by ~3.5°C than deeper water (Fig. 7A).

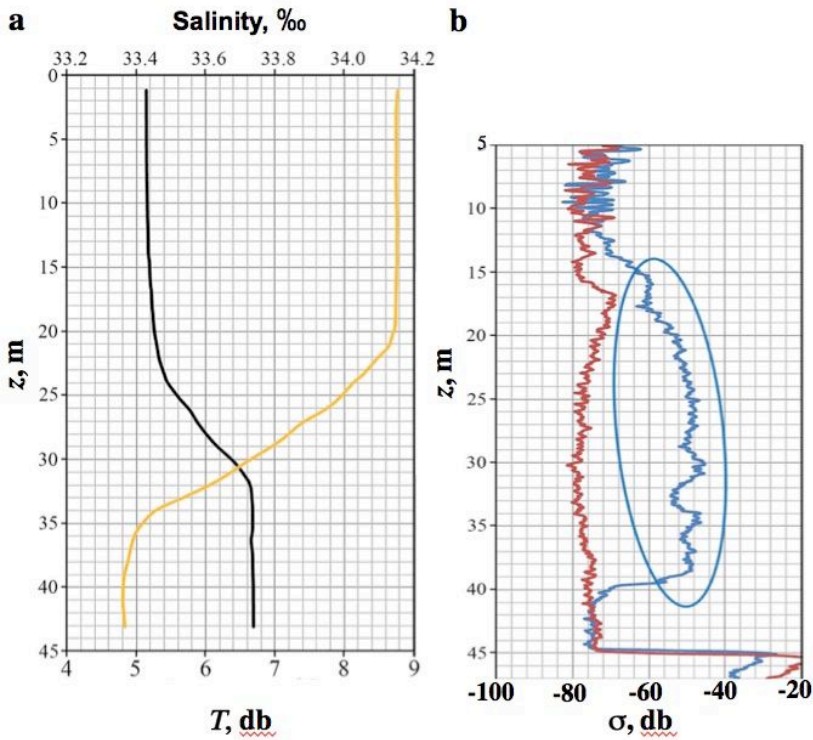


**Figure 7. a.** Salinity, and temperature, *T*, depth, *z*, profile during bubble plume calibration experiment in
the Laptev Sea. **b.** Single beam echosounder sonar return integrated across the plume, σ, with *z* for no
bubble plume (red) and a bubble plume (blue), bubble plume σ circled.
Bubbles have high density-contrast with water and thus are strong sonar targets that are distinguished
easily from the background (Fig. 7b). SBES data contains significant geometric uncertainty, which is
evident in the overlap in time of sonar returns for the calibration bubble plume (Fig. 8) and results from
current advection of the plume orthogonal to the page. MBES addresses this SBES deficiency. For



example, the SBES sonar loses the bubble plumes once they have entered the wave mixed layer, where
currents often shift, but the multi-beam sonar continues to follow them to 13 m depth, slightly below the
draft of the R/V *Viktor Buynitsky*.

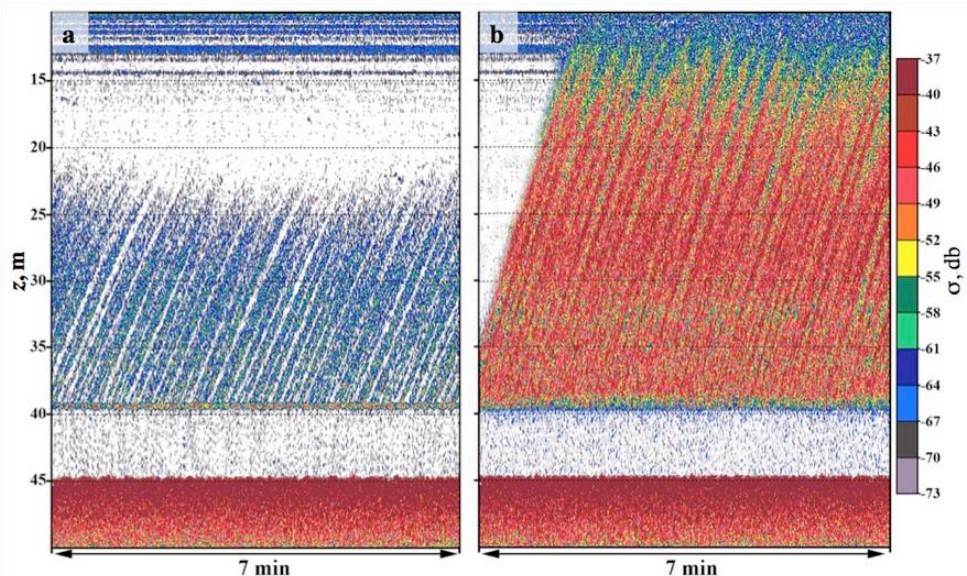


**Figure 8.** Plume-integrated sonar return, slume-icalibration bubble plume from 40-m depth, *z*, experiment
conducted for **a.** 0.042 L/min and **b.** 1.1 L/min at 5.5 bar for the single beam sonar.
Data analysis and visualization was performed with custom MatLab routines (Mathworks, Mass.) that
first geo-rectified each ping and then assembled the data for each experimental run into a 3-dimensional
array of depth, *z*, transverse distance, *x*, and along track distance, *y* (or time, *t*, if stationary).
Noise is the most common sonar return and was isolated from the bubble-plume signal based on setting a
threshold from the sonar return occurrence distribution, $\Psi(\sigma)$ (Fig. 9a). $\Psi(\sigma)$ showed a noise $\Psi(\sigma)$ at
approximately -80 db that clearly is distinct from the stronger, but less common, bubble $\Psi(\sigma)$ seen in Fig.
7b, for example. Based on inspection of $\Psi(\sigma)$, a noise threshold value of -70 db was selected, which
provided a 5-8 db transition between noise and bubbles (Fig. 9a, arrow). In addition, obvious sonar
artifacts, which can exhibit strong sonar return signatures, were masked by a swath constraint–i.e., spatial
segregation. Specifically, the plume center was identified at each depth-filtered to ensure continuity with
depth and only samples within a specified horizontal distance from the plume centerline that tightly
constrained the plume above the noise threshold were incorporated into the analysis.



For the calibration experiments, plumes with volume flux, $Q$, from 0.019 to 1.1 L/s were created and
observed by both SBES and MBES systems (Fig. 9). The contribution of bubble plume weak and strong
sonar returns were investigated by their signature in $\Psi(\sigma)$. Specifically, $\Psi(\sigma)$ was modeled by a piece-
wise least-squares, linear-regression analysis of $\Psi(\sigma) = a\sigma(z)^b$, which then was compared to expected
trends in plume evolution of a rising bubble plume. Fit parameters are shown in Supplemental Table S1,
with example data and fits for the 0.8 L/s plume shown in Figs. 9d-9f for three depth windows (all below
the WWML).

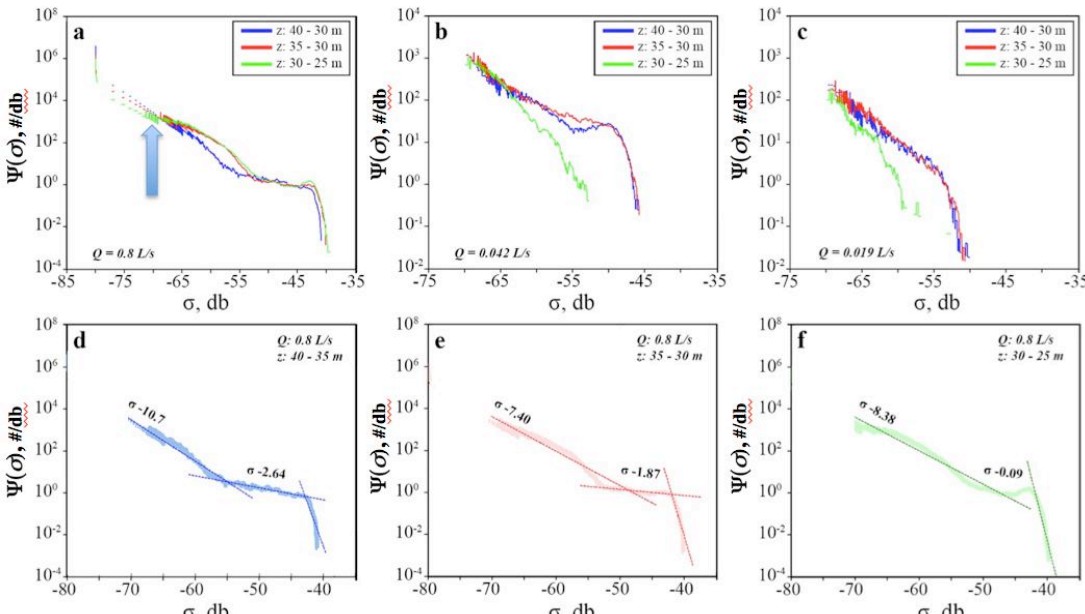


**Figure 9.** Plume-integrated sonar return, $\sigma$, occurrence, $\Psi$, normalized to sonar bin-width (sonar bins are
logarithmically spaced) for **a.** full water-column for a flow, $Q$, of 0.8 L/s – unthresholded for processed
depth windows, $z$, arrow shows noise threshold. $\Psi(\sigma)$ thresholded for **b.** $Q$ = 0.042 L/s, **c.** 0.019 L/s and
with linear fits for $Q$ = 0.8 L/s for **d.** $z$ = 35-40 m, **e.** 30-35 m, **f.** 25-30 m. Data key on figure. Fit
parameters in Supplemental Table S1.
$\Psi(\sigma)$ for low and high flows exhibited distinctly different characteristics with $\Psi(\sigma)$ for the intermediate-
flow plume exhibiting characteristics of both low and high flows. A weak sonar return represents small
bubbles, while strong returns may reflect large bubbles or it may reflect dense aggregations of small
and/or large bubbles. Thus, as a bubble plume rises, the relative importance of small bubbles should
increase as small bubbles disperse from the plume, spreading the weak sonar return over a larger volume.
The weakest flow plume shows a clear trend of a two-part power law at the deepest depth for $\Psi(\sigma)$ (Fig.





9c; Supplemental Table S1); however, $\Psi(\sigma)$ remained constant for both weak and strong returns as the
bubble plume rose (steepening increases the relative importance of weaker $\sigma$) to 30 m then steepening
abruptly, emphasizing smaller bubbles ($b$ = -8, -7, -12 for weak $\sigma$ for the 45-40, 40-35, 35-30 m depth
windows, respectively). For the weaker flows, 0.042 and 0.019 L/s (Figs. 9b, 9c), the strongest sonar
returns disappear completely at the shallowest depth, consistent with bubble-plume dispersion and bubble
dissolution.
The deepest depth window for the high-flow plume (Fig. 9d) exhibits a bi-modal $\Psi(\sigma)$ with stronger
returns more common relative to weaker returns than in the low flow plume (Fig. 9c) or than "predicted"
by extrapolating the power law ($\sigma^{-10.7}$) to the stronger returns (Figs. 9d and 9f, respectively). As this
plume rose, $\Psi(\sigma)$ for the weak $\sigma$ decreased in relative importance while $\Psi(\sigma)$ for stronger $\sigma$ remains
constant – the power law exponent, $b$, for the intermediate depth (-7.4) was less steep than for the deeper
(-10.7) and shallower (-8.4) depths. Thus, most of the evolution of $\Psi(\sigma)$ is due to a spatial expansion of
weaker $\sigma$ (i.e., smaller bubbles), while the strong $\sigma$ bubbles remain relatively uniformly constrained with
depth. The overall increase in $\sigma$ with rise is the same character observed in the precursor study (Fig. 6),
which featured strong flows comparable to those shown in Figs. 9d-9f.
$\Psi(\sigma)$ for the intermediate flow plume (Fig. 9b) shares characteristics of both the high and low flow plume
$\Psi(\sigma)$, bi-modal at the deepest depth with a pronounced strong $\sigma$ peak in $\Psi(\sigma)$ (like the high flow plume)
evolving into a dual power law as the plume rises (like the low flow plume $\Psi(\sigma)$. Thus, $\Psi(\sigma)$ for the
intermediate flow plume evolved through the patterns of the strong and weak flow plumes as it rose.
These plumes all originate from a point source and disperse as they rise, thus bubble-bubble acoustical
interactions should decrease as the bubbles rise. With the exception of the strongest plume, plume rise
decreases $\sigma$; however, for the strongest flow plume, rise initially increases return, similar to the trend in
the precursor study (Fig. 6), which was for comparably high flows, albeit over fewer depths much closer
to the source. Example MBES data are presented in the Supplemental Materials, Figs. S1 and S2.
The depth and flow dependent sonar calibration curves, $\sigma(Q,z)$, were derived to account for the depth-
evolution of bubble-bubble acoustic interactions as the bubbles rise (Fig. 10). Specifically, $\sigma$ above the
noise threshold in the spatially-segregated boxes in each depth window was averaged across 7-minutes of
sonar data to derive $\sigma(Q,z)$. The MBES and SBES calibration datasets show saturation at high flow,
similar to Greinert and Nützel (2004), which is evidence of bubble-bubble acoustical interaction. For high
flows, this likely includes sonar shadowing of more distant bubbles by nearer bubbles (decreasing total
return). At low flow, sonar return increases with increasing flow at a rate far faster than linear addition of



the number of bubbles–for a flow doubling ($Q$=0.02 to 0.04 L/min), σ should only increase ~6 db (
20$\log_{10}$(2) ), yet increases are much larger.

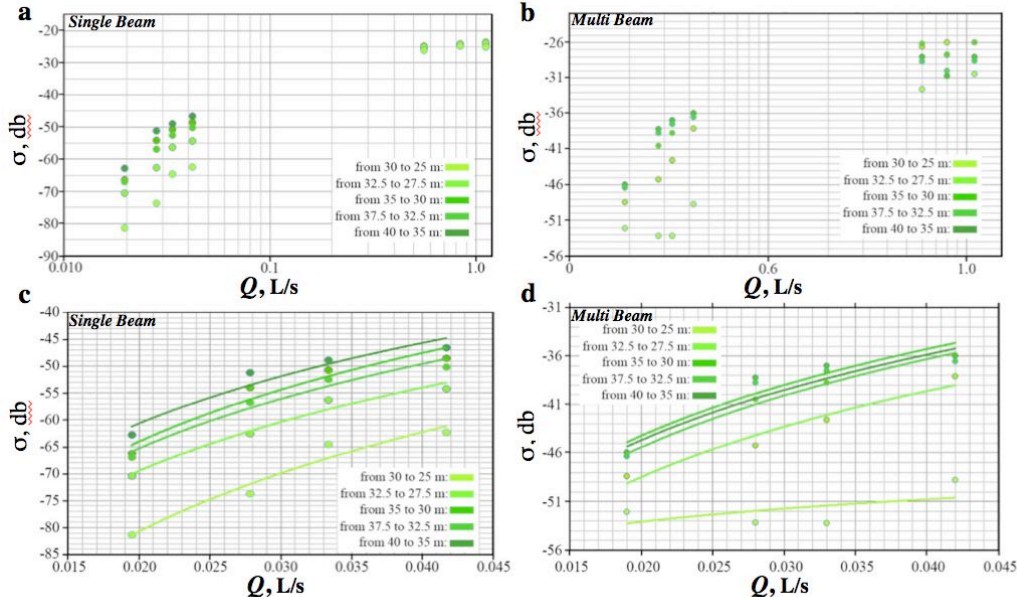


**Figure 10.** Sonar return, σ, with respect to volumetric flux, $Q$, calibration curves for the single-beam
sonar for **a)** all $Q$, and **c)** low $Q$, and for the multibeam sonar for **b)** all $Q$ and **d)** low $Q$. Fit parameters are
shown in Supplemental Table S2.
The calibration curves confirmed the existence of a depth dependency in σ for both SBES and MBES
systems (Fig. 10). For low flow plumes, σ decreases with rise and is non-linear with $Q$. In contrast, for
high flows, both SBES and MBES saturate or are near saturation although there is significantly more
variability in the MBES data. Close inspection of the high-flow plume MBES data revealed undulations,
which may have led to depth aliasing of the return in the 5-m depth windows. The high flow calibration
plumes are relevant for major seep bubble plumes such as in COP seep field; however, plumes in the
ESAS study area were not this strong, and the strong calibration plumes are not discussed further. In
contrast, the low flow calibration plumes are comparable to typical minor bubble plumes (Leifer, 2010)
and span the range of natural seepage observed in the study area.
These *in situ* calibration curves were derived for application to seep bubble sonar survey data, and
accounts for the vertical velocity of the bubbles, which includes the upwelling flow, which is itself
dependent on the flux (Leifer, 2010). However, the calibration curve should account for the difference in





depth between the seep study area and the calibration plumes (70 m versus 40 m) and in composition –
the seep gas primarily was methane, while the calibration gas was nitrogen. Both of these factors have
non-negligible implications for the bubble dissolution rates of the two different plumes. As a result, the
calibration should account for the differing dissolution rates and thus differing mean volume flux over the
depth windows.
**3.2. Bubble Dissolution Rates and Volume Flux**
Bubble dissolution or gas outflow for each gas species, $i$, is driven by the concentration difference, $\Delta C$,
between the bubble and the surrounding water,
$$F_i = k_{Bi} A (\Delta C_i) = k_{Bi} A (C_i - H_i P_i) \qquad\qquad (4)$$
where $k_B$ is the individual bubble gas transfer rate and depends on the gas diffusivity, $A$ is the bubble
surface area, $H$ is the Henry's Law equilibrium, and $P$ is the bubble partial pressure. To address the
difference in seep and calibration gases, a numerical bubble propagation model was used to explore the
relative dissolution rates for the two types of bubble plumes. The bubble model is described elsewhere
(Leifer et al., 2006; Leifer et al., 2015; Rehder et al., 2009). In brief, it solves the coupled differential
equations describing bubble molar content (Eqn. 4), size, pressure, and rise for each bubble size class in a
bubble plume.
Bubble size distributions, $\Phi$, for the calibration and seep bubble plumes were not measured, thus,
modeling provides a first attempt to quantify the biases that can be introduced. Implications of these
simplifying assumptions are discussed in Section 4.4. The model was initialized with a typical (Leifer,
2010) minor $\Phi$ (Fig. 11a) for either $CH_4$ or nitrogen bubbles, dissolved air gases at equilibrium in the
water column, the observed CTD profile (Fig. 11b), and a 10 cm s$^{-1}$ upwelling flow. This is an average
upwelling flow, which is too low for the highest calibration flow and too high for the lowest (Leifer,

2010).





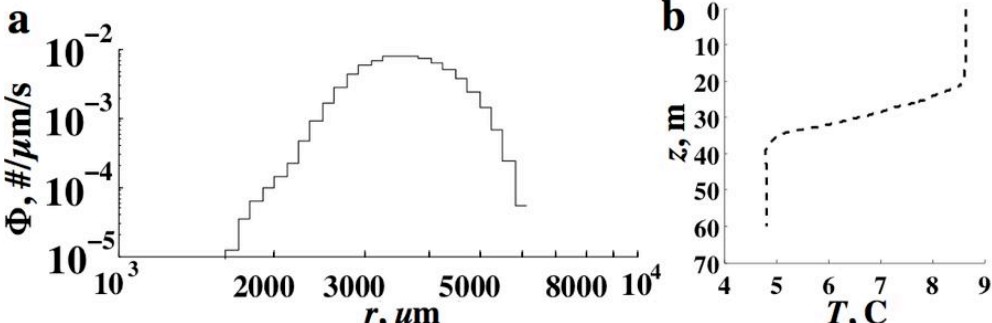


**Figure 11. a.** Minor bubble plume size distribution, $\Phi$, with respect to radiuss, $r$, used to initialize the

bubble model. **b.** Temperature, $T$, depth, $z$, profile used in model.

As a nitrogen bubble rises, it grows primarily due to decreasing hydrostatic pressure, and to a lesser
extent from oxygen inflow, while it shrinks from nitrogen outflow (Fig. 12). The numerical simulations
show that for the first three, 5-meter depth windows, the depth-averaged total bubble plume volume,
$<Q_z>$, increases (Fig. 12b) by 4.7%, 15%, and 29%, respectively.

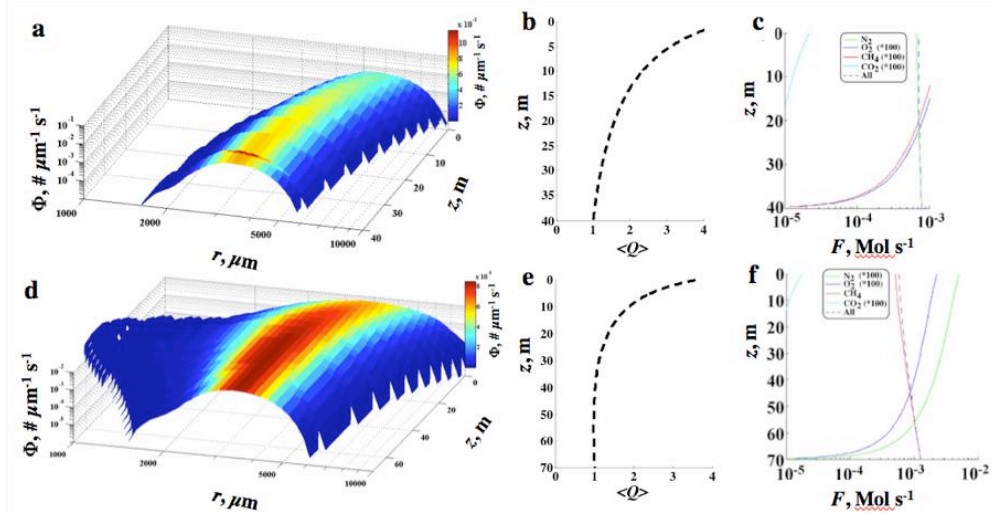


**Figure 12. a** Depth, $z$, evolution of the bubble plume size distribution, $\Phi$, for a nitrogen minor plume
(calibration) from 40 m and **d** for a $CH_4$ seep plume from 70 m. Seabed normalized volume averaged over
depth window, $<Q>$, of the rising bubble plume for **b.** calibration plume, and **e.** seep plume. Molar
vertical flux for **c.** calibration plume, and **f.** seep Data keys on panels.
In contrast to the 40-m nitrogen calibration bubble plume, there are dramatic changes in the size
distribution of a pure $CH_4$ minor seep bubble plume rising from 70-m depth with the smallest bubbles



dissolving and the largest bubbles growing (Fig. 12d). Overall, air uptake and decreasing hydrostatic
pressure largely balance dissolution for the plume overall for the first 50 m of bubble rise and $\langle Q_z \rangle$
remains roughly stable (Fig. 12e) – $Q$ decreases by 0.7%, 0.2%, and 0.0% in the first three 5-meter depth
windows, respectively. Note, stable $Q$ does not imply constant total $CH_4$ bubble content, which
continually outflows the rising bubble.
The volume correction factors between the calibration-plume and the seep plume are 0.948, 0.868, and
0.775 for the 65-70, 60-65, and 55-60 m depth windows, respectively. This shows that the calibration
plume $Q$ averaged over the 35-40 m depth window is ~5% greater than the seep bubble plume $Q$ for the
70-65 m depth window.

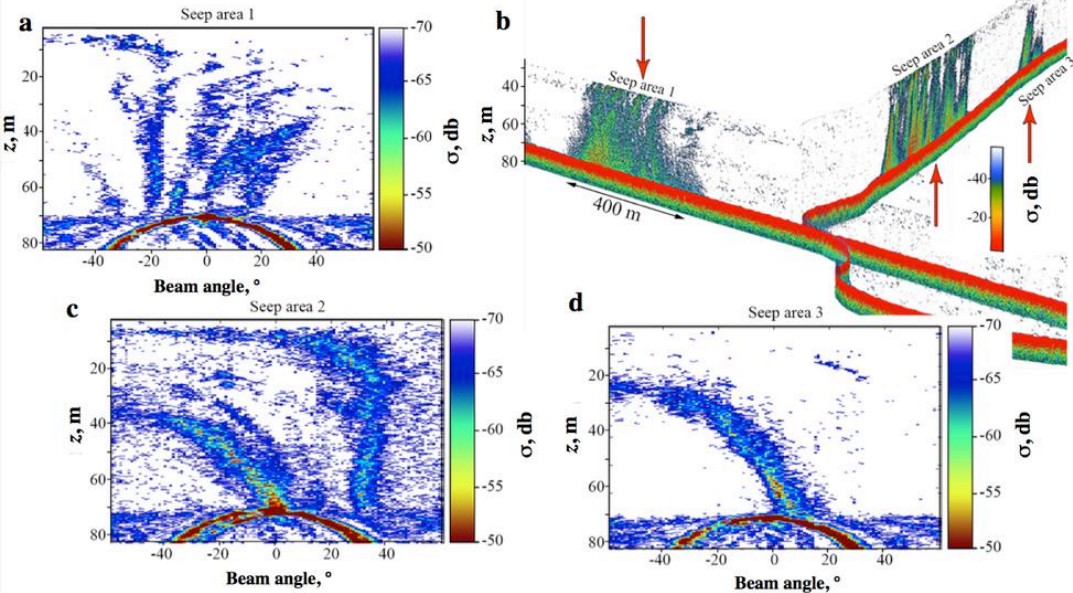

**Figure 13.** Sonar return, s, with depth, $z$, of seep bubble plumes in the Laptev Sea. **a. c. d.** Multibeam
sonar data, single ping, in each of the seep areas, locations labeled on b. **b.** Single beam sonar data. Size
scale and data key on panels.
**3.3. Natural Seepage Sonar Observations**
The depth-dependent calibration was applied to seep sonar data collected in the Laptev Sea for 70-m deep
water under conditions of strong currents (Fig. 13). Three seep areas were surveyed, two weak and one
strong, all with numerous plumes. The MBES data illustrates the additional spatial information missing in
SBES systems. For example, Seep Area 1 in the SBES data (Fig. 13b) appears to show extensive diffuse
seepage, which the MBES data (Fig. 13a) reveals is many low-flow discrete bubble plumes.

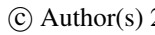



The flux for the seep areas (Fig. 14) was mapped by averaging the seepage flux in the 65-70 m depth
window in 1-m$^2$ quadrats after application of the calibration curves and correction factors. The deepest
depth window was chosen to preserve better the seabed location of emissions for spatial analysis.
Seep Area 2 was stronger than the other seep areas by an order of magnitude and clearly showed a
northeast-southwest trend, which also is apparent in all seep areas. Note, some of the striation patterns,
primarily of the weaker returns, are consistent with the very strong currents detraining small bubbles out
of the plume in the direction of the sonar beam fan. On a second, east-west leg, Seep Area 1 was surveyed
with currents not-aligned with the sonar beam fan and does not exhibit the striation. Further evidence of
this current effect is shown in the sonar ping data (Fig. 13a vs. Figs. 13c and 13d); where Seep Area 1
does not show the extreme tilt across beams as in sonar data for Seep Areas 2 and 3. Thus, the linear seep
trends must reflect geological control.

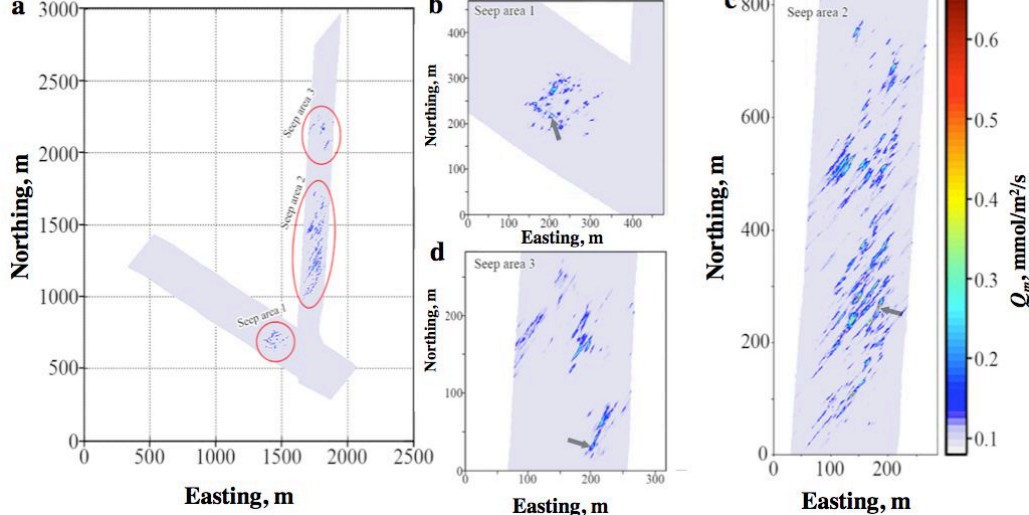

**Figure 14.** Seep mass flux, $Q_m$, occurrence, $\Psi(Q_m)$, normalized to flux bin-width (bin widths are
logarithmically-spaced) for **a** all seep areas, and for **b-d** Seep Areas 1-3 with power law fits. Data key on
panel a. Fits in Table 2.
Seepage spatial structure showed numerous seeps clustered around the strongest seep with an apparent
modulation at distances of ~100 m (Supp. Fig. S4). In seepage areas 1 and 2 the dominant seep plumes
were as strong as 0.3 mmol m$^{-2}$ s$^{-1}$ (7.4 cm$^3$ s$^{-1}$) while the dominant seep plumes in the stronger Seep Area
2 (Fig. 13c) released >0.6 mmol m$^{-2}$ s$^{-1}$ (15 cm$^3$ s$^{-1}$).


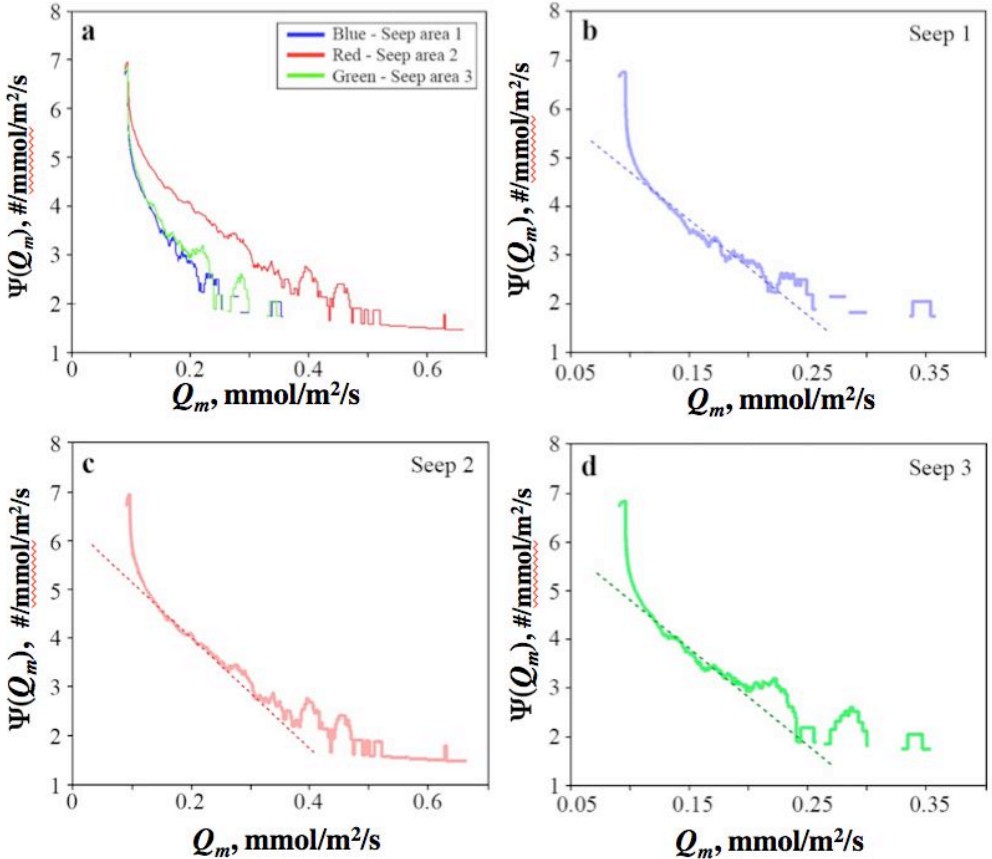


**Figure 15.** Seep mass flux, $Q_m$, occurrence, $\Psi(Q_m)$, normalized to flux bin-width (bin widths are logarithmically-spaced) for **a** all seep areas, and for **b-d** Seep Areas 1-3 with power law fits. Data key on panel a. Curve fits presented in Table 2.

The mass flux, $Q_m$, occurrence distribution, $\Psi(Q_m)$, was calculated for each seep area and showed Seep Area 2 contained the largest number of strong seep plumes followed by Seep Area 3 and then Seep Area 1 (Fig. 15). For these seep areas, $\Psi(Q_m)$ for weak emissions asymptotically approached ~0.1 mmol/m²/s (2.5 cm³/s)–the noise level. Thus, the calibration flows (Fig. 10) bracketed from the MBES noise level to the largest observed seep plume. Seep Area 2 exhibits both greater fluxes and a shallower power law (Fig. 15c). Furthermore, all three seep areas exhibited positive anomalies or peaks in $\Psi(Q_m)$ for stronger flux seepage. These peaks signify a preferred emission mode–i.e., multiple seeps with similar emission fluxes.

For weaker seeps with good signal to noise ($Q_m > 0.15$ mmol/m²/s), the power law fits are nearly identical, 6.65, 6.27, 6.80 (Table 2) for Seep Areas 1, 2, 3, respectively. Total flux in each seep area was determined



by area integration and was 5.56, 42.73, and 4.88 mmol/s for the MBES data (Table 2). SBES-derived
emissions were biased lower compared to MBES, by 3.7% - 36% for the seep areas, with best agreement
for Seep Area 2.
TABLE 2 HERE
**4. Discussion**
**4.1. Bubble-Bubble Acoustic Interaction**
We presented results of an *in situ* experiment to investigate the evolution of bubble plume sonar return
from rising engineered bubble plumes spanning two orders of volume flow rates from flows that were
comparable to typical minor plumes and very strong major plumes at the high end (Leifer, 2010).
Calibration plume sonar return increased strongly and non-linearly with flux, ~15 db for a flow doubling
from 0.02 to 0.04 L/s. This increase is faster than the 6 db increase that would be expected by simply
summing the sonar cross sections of the doubled number of bubbles. Instead, the increase suggests strong
bubble-bubble acoustical interactions. Specifically, with increased flow, overall plume dimensions
expand more quickly, leading to less bubble shadowing and shallower sonar occurrence slopes at the
same height above the nozzle (Fig. 10). In contrast to the overall plume dimensions, which include
smaller more dispersed bubbles, the dense core of large bubbles tends not to disperse and is largely
insensitive to height (Fig. 9). Thus, for the dense plume core, increased flux increases bubble shadowing
such that the signal of the additional bubbles is blocked by other bubbles and sonar return becomes nearly
independent of flow, i.e., saturated (Figs. 10a and 10b). Greinert and Nützel (2004) observed similar
behavioral regimes for air bubble plumes in far shallower water. Thus, the calibration bubble plume
provides strong evidence of non-negligible bubble-bubble acoustical interaction at both low and high flow
rates. Furthermore, the non-linearity of the relationship is shown by the relationship between σ and *Q* as
the bubble plume rises and disperses. Thus, bubble-bubble acoustic interactions remain significant even
after the plume has risen 15 m.
As high-flow bubble plumes rise, the weak sonar return portion of the plume evolves due to small bubble
dispersion, leading to an increase in the integrated sonar return (Fig. 9), a pattern observed for the air
calibration experiment in the Coal Oil Point (COP) seep field (Fig. 6). In the COP seep field study,
calibration flows extended from comparable to far higher flows than those reported herein, and found that
the depth-dependent sonar return increased with height on finer depth scales (Fig. 6) than obtainable in
the Arctic experimental configuration. This is interpreted as due to decreased bubble "shadowing" of
more distant bubbles as the plume expands. In the case of the ESAS calibration flows, the depth evolution



of the sonar occurrence distributions showed a strong influence from small bubble dispersion leading to
an expansion of the plume volume and an increase in integrated sonar return.
As the low-flow calibration plumes rise and disperse, sonar return decreases. Overlapping intermediate
depth windows were evaluated and confirmed this was not an artifact of plume oscillatory motions
aliasing the return signal across the depth windows. The decrease in overall sonar return with rise is (by
definition) a decrease in scattered sonar energy. This suggests that greater energy scatters back to the
sonar when the plume is spatially denser.
**4.2 Bubble Detrainment and Bubble-Bubble Acoustic Interaction**
The current artifact striations in the natural seep sonar data are consistent with the importance of bubble-
bubble acoustic interaction. Specifically, where seep bubble plumes were imaged under high currents,
small bubbles were advected out of the plumes into the downcurrent water in the beam fan, and observed,
but not when the beam fan was perpendicular to the currents. In the case of the beam-fan-current co-
orientation, scattered acoustic energy interacts with nearby downcurrent bubbles, which are in the beam –
the cross-track beam is broad (120°), while the along-track beam is narrow. In contrast, when cross-
oriented, the sonar beam fan fails to image the detrained bubbles. This provides clear evidence of bubble-
bubble acoustic sonar interactions for distances larger than the plume dimensions.
**4.3. Weak and Strong Sonar Bubble Contributions**
The sonar occurrence distributions were bimodal for intermediate and strong calibration plume.
Specifically, weak sonar returns were well described by a steep power law ($\sigma^{-b}$, $b > 7$) for high $\sigma$, and for
all but the weakest plumes, a gently sloped "shelf," which in the intermediate flow case ($Q = 0.42$ L/s)
was negative leading to a second peak. These changes mimic those observed in how the bubble size
distribution changes with increasing flow reported in Leifer and Culling (2010). In that study, with
increasing flow, the low flow plume bubble distribution is a narrow Gaussian (minor bubble plumes),
which shifts to a power law at high flow (major bubble plumes) that spans small to very large bubbles.
Intermediate flows exhibit characteristics of both. A steep power law implies spatial constraint–i.e.,
bubbles are not dispersing and creating returns across a spectrum of strengths. Furthermore, the
occurrence of strong return is invariant with depth for the strong plume (Fig. 9a), indicating that the large
bubble core of the strong plume remains spatially constrained. At intermediate flows, the sonar
occurrence distribution infills–creating a shallow power law for higher flows and stronger returns. In
contrast, turbulence and currents tend to disperse small bubbles, which are present in both weak and
strong flow bubble plumes. As the plume rises, the signal from these bubbles eventually become lost to



dissolution and dispersion that reduces sonar return signal to the noise level, leading to a steepening of the
sonar return occurrence distribution for weak returns.

### 4.4. Bubble Size Distribution

Bubble size distributions have been reported for other Arctic seep sites (Shakhova et al., 2015), but
equipment to make such measurements were unavailable for this study. Low flow seep plumes are termed
minor (Leifer, 2010) and are well described by a Gaussian function (Leifer and Culling, 2010). With
increasing flux, the peak radius of the plume increases, until a critical flux above which the bubble size
distribution becomes more complex, until eventually being described by a power law, termed major
(Leifer and Culling, 2010). The transition from minor to major depends on sediment characteristics and
physical oceanographic conditions such as temperature and salinity that affect bubble plume formation
(Asher et al., 1997; Haines and Johnson, 1995).
Bubble modeling was used to address the effect of evolving bubble size distribution with flow in
application of calibration air or nitrogen (preferred for safety reasons) bubble plumes to seep bubble
plumes. In this study, we applied a first approximation using a typical minor bubble plume size
distribution. Clearly initializing the model with measured plumes would improve the accuracy of the
volume correction factor and hence sonar-derived flux. Still, the primary goal in our study is to
demonstrate with a simple approximation that bubble size matters and should not be neglected.
Although the simulations were conducted to correct between a nitrogen calibration plume and pure
methane seep bubbles, a compressed air calibration plume could be used and would behave highly
similar. If the seep bubbles contain other non-trace gases, their outgassing could impact significantly
bubble size evolution with rise. This is particularly relevant for a gas like carbon dioxide, which is far
more soluble than methane, and thus can lead to rapid bubble size change in the deepest depth windows.
Note, bubble dissolution is strongly depth dependent (Leifer and Patro, 2002a) and thus the greater the
depth discrepancy between calibration plume and seep plume, the larger the correction factor.
The sonar return of a bubble depends on its size (and shape which depends on size) and relationship to
flux also depends on the vertical rise rate including upwelling flows. Thus, future studies should
investigate both the size distribution and upwelling flow for a range of flow rates.

### 4.5. Field Comparison of MBES with SBES

The MBES and SBES systems were calibrated with the same nitrogen gas bubble plumes, thus the two
systems should agree in terms of flux observations. Calibration flows spanned very weak flow ($Q = 0.19$
L/s) to very strong flows ($Q = 1.1$ L/s). The low flow calibration bubble plume was below the seep field





noise floor of the MBES system, while the high flow was more than an order of magnitude greater than
field observations.
Field observations showed far better agreement between systems for Seep Area 2 than the other seep
areas (Table 2). This most likely relates to the greater relative importance of stronger seeps that are well
above the noise level relative to the other seep areas. The calibration flows (Fig. 10) showed that SBES
sonar return was weaker for the same flow than the MBES sonar return. Geometric uncertainty likely
played a role in the SBES downward flux bias.
**4.6. Seepage Spatial Characterization**
The seepage spatial and strength distribution in the ESAS (Fig. 14) share similarities with structures in the
COP seep field (Fig. 1). Subsurface geologic structures control the seepage spatial-flux distribution by
creating the pathways through which seepage migrates to the seabed and ocean - seepage areas must
occur where geologic structures allow. In the COP seep field, strong seepage areas are located at
intersecting non-compressional faults and fractures (Leifer et al., 2010). Furthermore, these faults and/or
fractures themselves are preferred migration pathways that connect subsurface reservoirs to the seabed,
with seepage tending to manifest along their trend (Leifer et al., 2010).
In the ESAS seepage map (Fig. 14), two spatial trends were manifest, one northeast-southwest of
individual vents and second a north-south elongation in Seep Area 2. Both trends were aligned with the
two weaker seepage areas. Furthermore, the northeast-southwest trend is apparent within Seep Area 2.
Here, fractures in submerged permafrost could play a similar role to the role of fault intersections in the
COP seep field; however, more extensive seep area mapping is needed for validation, and/or penetrating
sonar data that can image near surface rock strata. On smaller length scales, there is an evident striation
pattern that defines vent locations suggesting a subsurface linear geological control on meter length
scales.
High flow seepage requires high permeability migration pathways, while low flow seepage occurs along
low permeability migration pathways if the driving pressure between the deeper reservoir and the seabed
is constant across the active seepage area (Leifer and Boles, 2005). Thus, the stronger and more numerous
and extensive seepage emissions from Seep Area 2 indicates higher subsurface permeability and
subsurface connectivity with more numerous migration pathways than the other seep areas (Fig. 14).
Seepage connectivity can be envisioned topologically as an inverted branched structure (Leifer et al.,
2004) where central stronger seepage is surrounded (generally) by weaker seepage (Supp. Fig. S4). Given
that permeability is inversely related to resistance in the migration pathways, stronger seepage is fed by
migration along pathway(s) with lower resistance (higher permeability), while weaker seepage is fed by



migration along pathways with stronger resistance (lower permeability). One implication of a range of
migration pathways with different resistance is that lower resistance seepage adjusts to changes in
seepage easier than higher resistance seepage – thus strong seeps become stronger, while weak seeps are
more likely to activate/deactivate with changes in emissions (Boles et al., 2001; Bradley et al., 2010). The
balance between seepage emissions for different migration pathways with a range of permeability
underlies the flux occurrence distribution (Fig. 15).
The mapped seepage emissions demonstrated highly similar geologic spatio-flux control. Specifically,
weak seepage flux exhibited a fractal dimension, $b$, of -6 (Fig. 15), which characterizes how seepage
distributes itself between high and low permeability migration pathways. Note, the actual power law
likely is slightly exaggerated due to bubble detrainment into the beam fan in Seep Areas 2 and 3, which
spreads sonar return spatially; however, Seep Area 1 does not have this beam fan effect, yet exhibited a
similar $b$ to the other areas. This argues that the shallow seabed structure (fracture, porosity, etc.) related
to low permeability migration pathways is common across the areas, with the main controlling factor
being the number of bubbles escaping per second per unit area of seabed.
This power law does not extend to the largest seep fluxes, which manifest as perturbations (peaks) above
$b = -6$ power law in the flux occurrence plot. Higher flow plumes, and thus high permeability pathways,
could represent a failure of the normal seabed structure (that governs the weak seepage) from stresses
and/or talik melting, leading to focused high flow migration pathways that help define where the seep
areas lie.
In the Arctic, subsea permafrost degradation from heating both below (geologic – most strong in faulted
zones) and above (riverine inputs and overall Arctic Ocean warming) creates migration pathways that
manifest as seep spatio-flux distributions. The presence of active seepage in this region likely relates to
these heat flows, with the hotspots likely related to taliks and/or subsea thaw lakes, whose locations are
controlled by linear geologic structures. In the ESAS, grabens are often linear structures, which often are
correlated with paleo-river valleys, and could also cause small-scale co-aligned fractures that lead to
seepage being along linear trends. The similarity in the emission probability distribution power law ($b$=-6)
indicates that subsurface permeability exhibits a fractal distribution that is similar between the three areas
– arguing for similar formation mechanism, i.e., taliks. In this case, at the intersection of the two linear
trends, where migration is higher and thus heat flow likely is higher, talik evolution would be greater,
leading to more higher permeability migration pathways.



**4.7. Broader Implications**

There are enormous carbon stores sequestered in marine-permafrost in the Arctic, which are of particular concern for release as the warming Arctic Oceans transfer heat faster than from the atmosphere to terrestrial permafrost. Migration from this submerged permafrost reservoir to the ocean has created a vast marine seep field that lies entirely in shallow waters with emissions contributing directly to atmospheric budget (Shakhova et al., 2014). Widespread ESAS seabed bubble emissions have been documented (Shakhova et al., 2014, 2015), demonstrating failure of the permafrost's integrity and making methane and additional organic carbon available for microbial methane generation.

The observations support the hypothesis that the current state of sub-sea permafrost is a controlling factor to the spatial variability in methane seabed fluxes, and is undergoing destabilization from warming (Shakhova et al., 2010a, 2010b). The current state of subsea permafrost beneath the ESAS is a potential key to understanding whether and how, methane preserved in seabed reservoirs, escapes to atmosphere (Shakhova et al., 2009a, b, c; Shakhova et al., 2010a, b). Currently our state of knowledge engenders enormous uncertainty in future emissions in large part due to the paucity of data (Shakhova et al., 2014, 2015). Among the new tools and techniques needed to evaluate these fluxes quantitatively over wide areas, *in situ* calibrated sonar shows significant promise.

**4.8. Future Directions**

In this study, bubble plume spanning almost two orders of magnitude, from 0.019 to 1.1 L/s were engineered; however, a key intermediate range (0.045-0.8 L/s) was missed. This is the regime where bubble plumes shifts from a non-linear relationship between sonar return and flow to saturation where sonar return is largely independent of flow. Furthermore, experiments should follow the plume for more than 15 m; however, currents made this infeasible. Also, the calibration plumes looked at isolated bubble plumes; however, seep bubble plumes often escape from nearby vents into plumes that eventually merge. Given the importance of bubble-bubble acoustic interactions, calibration studies should compare the same total flux from one to several closely located bubble sources to investigate whether there is convergence between single bubble plumes and multiple bubble plumes with rise height as the plume merge. Finally, studies in calmer waters could elucidate better the importance of small bubbles versus large bubbles to overall sonar return.

This study featured the novel use of a numerical bubble plume model to correct for different size evolution between calibration gas bubble plumes and seep bubble plumes. Uncertainty arises from the bubble size distribution, which needs to be measured for the calibration and seep bubble plumes at



multiple flow rates. Our approach was a simplified first effort with room for improvement, including
measurement of bubble size distributions in the field.

## 5. Conclusions

In this study, using the calibrated multi-beam and single-beam sonars we improved our ability to map and
quantify the methane release from seepage in the Laptev Sea outer shelf where subsea permafrost is
predicted to be mostly degraded. We created engineered bubble plumes in situ from 40-m depth spanning
almost two orders of magnitude, from 0.019 to 1.1 L/s. Non-linear curves relating sonar return to flux for
a range of depths demonstrated significant bubble-bubble acoustic interactions – precluding the use of a
theoretical approach of scaling bubble sonar cross section by the size distribution. Analysis of the depth
evolution of the bubble plume sonar occurrence for different fluxes found weak sonar return was well
described by a power law that likely correlated with small bubble dispersion, while strong sonar returns
were largely independent of depth, consistent with a central core of focused large bubbles. As a result,
plume sonar occurrence was bimodal for all but the weakest seepage.
The *in situ* calibration curve was applied to a natural seepage area from 70-m depth after accounting for
the different volume evolution of the nitrogen calibration plume and the methane seep bubble plume
through use of a numerical bubble plume model initialized with a typical (assumed) minor bubble plume
size distribution. The bubble model suggests ~5% difference between the calibration and seep plumes
over the first 5-m depth window. Three nearby seepage areas with total emissions of 5.56, 42.73, and 4.88
mmol/s from multibeam sonar data were mapped, with good to reasonable agreement (4-37%) between
single and multibeam sonar, although single beam emissions were biased lower. Seepage occurrence was
bimodal, with weak seepage occurrence in each seep area well described by a power law. This was
interpreted as suggesting primarily small minor bubble plumes, while a few stronger seepage plumes were
mapped that could be major plumes. Seepage mapped spatial patterns suggested subsurface geologic
control along linear trends.

## 6. Acknowledgements

We thank the crew and personnel of the expedition onboard research vessel *Victor Buinitsky*. We would
like to acknowledge financial support from the Government of the Russian Federation (grant #14,
Z50.31.0012/03.19.2014), the Far Eastern Branch of the Russian Academy of Sciences (RAS). At
different stages work was supported by the US National Science Foundation (OPP ARC -1023281), the
US National Oceanic and Atmospheric Administration (Siberian Shelf Study), Russian Foundation for
Basic Research (grants #13-05-12028 and 13-05-12041), and Headquarters of the Russian Academy of



Sciences (Arctic Program led by A.I. Khanchuk). N. S. and D. C. acknowledge the Russian Scientific
Foundation (grant #15-17-20032).

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

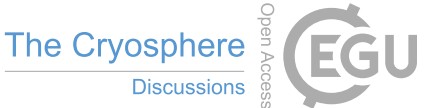

**Tables**

**Table 1.** Integrated depth-windowed methane flux estimates.

| Designation | $Q_{m\text{-SBES}}$* | $SQ_{m\text{-SBES}}$ | $Q_{m\text{-MBES}}$** | $SQ_{m\text{-MBES}}$ | Area | $U$ | $SQ_{m\text{-MBES}}$ |
|---|---|---|---|---|---|---|---|
| | (mmol/m$^2$/s) | (mmol/s) | (mmol//m$^2$/s) | (mmol/s) | (km$^2$) | (%) | (L/s) |
| Seep 1 | 0.22 | 3.78 | 0.33 | 5.56 | 0.017 | 32 | 0.14 |
| Seep 2 | 0.59 | 41.16 | 0.61 | 42.73 | 0.070 | 3.7 | 1.07 |
| Seep 3 | 0.26 | 3.96 | 0.33 | 4.88 | 0.015 | 19 | 0.12 |

$Q$ is volume flux, $Q_m$ is mass flux, U is uncertainty, where $U=(Q_{m\text{-MBES}}-Q_{m\text{-SBES}})/Q_{m\text{-MBES}}$

*SBES – Single Beam Echosounder, 65-70 m, depth window.

**MBES – Multibeam Echosounder, 65-70 m, depth window.

**Table 2.** Fit parameters for seep area flux occurrence.

| Name | $Q_{m\text{-1}}$* | $Q_{m\text{-2}}$ | a | b | $R^2$ |
|---|---|---|---|---|---|
| | (mmol/m$^2$/s) | (mmol/m$^2$/s) | (-) | (mmol/m$^2$/s) | |
| Seep Area 1 | 0.1 | 0.2 | -19.53 | 6.648 | 0.836 |
| Seep Area 2 | 0.1 | 0.3 | -11.34 | 6.27 | 0.9228 |
| Seep Area 3 | 0.1 | 0.2 | -19.85 | 6.798 | 0.8258 |

Fit from $Q_{m\text{-1}}$ to $Q_{m\text{-2}}$, where $Q_m$ is the mass flux rate