# Peer review of "Sonar Gas Flux Estimation by Bubble Insonification: Application to Methane Bubble Flux from the East Siberian Arctic Shelf"

_The Cryosphere, 2016_

## Referee Comment (RC1) · Anonymous Referee #1 · 9 Oct 2016

The manuscript aims to contribute to an improved understanding of methane emission from the East Siberian Arctic Shelf (ESAS), and presents results from investigations and analysis of engineered bubble fluxes and in situ observed bubble fluxes. Quantification of such fluxes can be made using acoustic techniques and devices, however, there are challenges associated it.

For instance, significant bubble-bubble interactions, which require the derivation of a calibration curve, as this interaction is not captured by theoretical calibration functions. Therefore, such calibration curves are derived for engineered (nitrogen) bubble plumes, and are then applied to in situ observed (methane) bubble plumes, for which results are reported and discussed.

[Figure]

While I believe quantification of gas fluxes from the ESAS to be a topic of interest and relevant to the audience of the SI " Climate–carbon–cryosphere interactions in the East Siberian Arctic Ocean: past, present and future" in "The Cryosphere", I can not recommend publication of this manuscript in its present form. The manuscript needs major revision before it can be considered for publication. This is mainly because the material is not presented in a sufficiently streamlined and focused way, making it difficult for the reader to follow. Some equations introduced lack complete and concise description, and many figures come in too low resolution and with insufficient explanations in the figure captions. Together, this implies that its is difficult to assess whether results and conclusions reached are substantial, and whether their interpretation and discussion is sufficiently complete.

Therefore, I recommend major revision of the manuscript, and include below some specific comments that might be of help during a revision.

The introduction comprises extensive subsections on Arctic methane and climate change, Study motivation, Marine seepage, Seep bubble measurements, and Sonar seep bubble measurements and is too broad, too long and too general to serve as a concise introduction to the topic presented in the manuscript. In view of the geographical focus of the SI, the inclusion of the pre-study carried out in the coastal waters of California would deserve a better justification, or could be removed in the interest of streamlining and shortening the manuscript. Equations (1) and (2) presented in Sect. 1.4. are ambiguous and not understandable from the information given in the text.

The methodology section briefly describes the three field-work moments (including field sites, weather conditions, technical description of experiment-set up) on which this manuscript is based: a pre-study in coastal waters of California, calibration experiments in the Kara Sea, and the major field campaign in the Laptev Sea. Given the contents of this section, a more natural label for this section would be e.g. "Field sites" or, "Field sites and experimental set-up".

[Figure]

However, a new methodology section could be introduced, where modeling and visu-
alization methodologies could be discussed, and example of which are now found at
eg lines 370 and onward, and lines 456-472 (that is, part of Sec. 3.2, containing now
the description of the numerical bubble plume model, the use of which is necessary to
account for differences in the engineered nitrogen plumes and the measured methane
plumes).

In the results section, numbering of subsections is incorrect: there are e.g. two sub-
sections labelled 3.1, the first one referring to results from the pre-study, the second
one to results from the calibration tests.

Parts of what is contained in the results section does not belong there thematically. For
instance, the first paragraph of (the second) Sect. 3.1 (lines 350-255) does rather be-
long to the "field sites and experimental set-up" section. Lines 360-366 do not describe
results, neither do lines 370 until ca 374. Lines 395-399, 420-424, and 447-454, e.g.,
seem more of discussion nature.

Results presented e.g. in Fig 7. lack complete description as no information regarding
the geographical location of the profiles is given. If the specific geographic location ins
not important, because the profiles are considered representative for a larger ares, then
this should be stated. Do both panels in Figure 8 show data acquired with an SBES?
In the text, mention is made that MBES addresses deficiencies related to geometric
uncertainty when using SBES- could a figure be added that shows SBES vs MBES?
Does Figure 9 show data from SBES or MBES, cf. also line 383? F

The discussion section is very extensive, with 8 subsections including discussion of
broader implications and future directions. Similar to the introductions, the discussion
section would benefit from shortening and streamlining, and a clearer link to the central
results presented (Figures 13 and 15?).

Figures: Throughout the manuscript, figures appear to have too low resolution. Figure
captions are often not explanatory and specific enough. For instance: What specifically

is presented in Fig 1.b? What do the colors mean? Does "red" indicate bubbles/seeps? What is shown in Fig 5? Existing annotations are hard /impossible to read and make understanding difficult.

---

## Referee Comment (RC2) · Anonymous Referee #2 · 12 Oct 2016

The East Siberian Arctic Shelf (denoted in the paper as ESAS) produces a huge amount of methane through the degradation of the permafrost on the shallow sea bed. The importance of this process for the global clime change is substantiated in the thoroughly "Introduction" accompanied with appropriated references.

The direct estimation of the methane flux in the water and in atmosphere in this region is of great importance. The well known sonar surveys are without a doubt effective method for such measurements. The authors apply this approach to the sonar data obtained with single beam echosounder (SBES) and multi beam echosounder (MBES) with high working frequencies 200 kHz and 260 kHz respectively. Instead the "theoretical" calculation of the volumetric flux Q transported by ascending bubbles based on the

back scattering strength $\sigma$ of the bubble plume, bubble size distribution (BSD) and rise speed of the bubbles (Weber et al., 2014 and additional ref. [1,2] bellow) the authors use the direct calibration of the sonar systems. Calibration curve was obtained using artificial bubble jet of nitrogen and demonstrates non-linear connection between gas flux Q and back scattering section $\sigma$ of the bubble jet (denoted in the paper as "sonar return") and a significant differences at the different layers of the bubble plume.

Calibration was carried out for both SBES and MBES echosounders. It is not clear how it was made for multi-beam system. Was it made for one vertical beam only or for several beams near the vertical direction or for all 128 beams in the fan in the resolution cells corresponding to the equal depth?

The authors try to explain the properties of the calibration curve with help of the "bubble-bubble acoustical interaction". This term appears already in the abstract and than pass as a "red thread" through the whole text of the paper. Corresponding to (Leifer and Tang, 2007) the "bubble-bubble acoustical interaction" is manifested as a shift of resonant frequency if the bubbles are disposed at distances less than 20*rb. In my opinion this effect does not play the role in the scattering, because in the broad BSD always are the bubbles with the resonant radii corresponding to the frequency of the echosounder.

Certainly the effects of multiple scattering, as it is described in "Discussion" (lines 544-561), determine the properties of the acoustical back scattering of the bubble plume. For example, the absorption of the acoustic energy in the over part of the plume (acoustical shadowing) due to the very small bubbles resonating on the high transmitted frequency can reduce significantly the sonar return of the underlying parts of the plume. But, in my opinion, the qualitative argumentation of the authors concerning the back scattering from the bubble plume is not convincing. For the better understanding of this problem the deep quantitative theoretical consideration is needed.

It is also not clear why the paragraph "COP seep field precursory study" (lines 269-284

and 334 – 348) is included in the paper. Of course, this investigation is very interesting as itself, but it does not contribute to the better understanding of the main scope of the paper. In this case the plume was insonified with the fan of beams from the side, but in the ESAS sonar survey and calibration – from the above. Furthermore, the fig.6 is difficult to understand and the axis "$\sigma$, dB" contains the values 102-106 in contrast to the negative values for "$\sigma$, dB" in all other figures.

The section "3.2. Bubble Dissolution Rates and Volume Flux" (lines 455-495) contains the brief description of the numerical model of the BSD transformation with the depth due to the gas exchange between bubbles and surrounding water. The authors write that it is necessary to disseminate the results of the calibration (obtained with nitrogen) on experimental data obtained with a different gas (methane) and at other depths. But it is not clear from the text of the section how it was made.

The headline of the paper reads: "Sonar Gas Flux Estimation by Bubble Insonification: Application to Methane Bubble Fluxes from the East Siberian Arctic Shelf Seabed" and the reader is entitled to expect a presentation of the results, covering significant part of the ESAS.

Really, the fig.4 demonstrates three polygons of the research cruise covering a big region of the Laptev Sea. But, from the other hand, only small regions with plumes are represented on the fig.14. The position of this area on the whole ship route is given neither is the text nor on the figure 14. Does it mean that only this plumes area were discovered?

Some technical remarks. At first, authors use the term "occurrence" for the histograms $\Psi(\ldots)$ of the sonar return $\sigma$ and gas flux in the plume Q. I think, "histogram" will be better for the understanding. Second, the explication to the Minnaert equation (lines 234 and 235) must be corrected.

I think that the paper may be published after clarification of certain points in the text and correcting technical errors.

1. S. I. Muyakshin and E. Sauter. The Hydroacoustic Method for the Quantification of the Gas Flux from a Submersed Bubble Plume // Oceanology, 2010, Vol. 50, No. 6, pp. 995–1001.

2. M. Veloso, J. Greinert, J. Mienert, M. De Batist. A new methodology for quantifying bubble flow rates in deep water using splitbeam echosounders: Examples from the Arctic offshore NW Svalbard //Limnology and Oceanography: methods. 2015, v.13, pp.267-287.
* * *

---

## Author Comment (AC1) · 26 Jan 2017

Both reviewers suggested that the manuscript could use from structural re-organization and a thorough proof reading, as well as shortening. We agree and have very significantly improved the manuscript. Specifically, We moved details of the modeling to the methodology section, as well as the calibration. We also moved up and improved the study motivation. These two efforts improved the manuscripts. Prior to this, we went through the entire manuscript on a line by line basis to improve readability and add definitions.

While the concepts of bubble-bubble interactions described in equation 4 are key to the article's main point, the equation was a distraction that has been replaced by a short

discussion, given that there is no effort to apply the equation (and its main point is that our understanding is too poor to even attempt to do so).

From the point of view of sonar observations, the bubble differential equations of radius and mass change are now presented with a sentence or two describing each. These equations are important also for the bubble model.

Addressing also the comment of Reviewer 1 and Reviewer 2, a better explanation for the scoping study is now presented in the revised study motivation, which is earlier in the manuscript. Also, the bubble model description has been moved to earlier, and a better explanation of the role the bubble model plays in the analysis ties in the bubble behavior equations has been added. Additionally, the differential equation for bubble size has been added, which ties changes in mass flux to size – i.e., what sonar observes, including explanation of the terms.

New section titles all of which are correctly numbered.

The number of figures in the main manuscripts has been reduced by two.

Reviewer 1: We apologize for the lack of geographical information for Fig. 7, but this was part of the agreement necessary to have permission to release these data (a small subset of the overall data). These data were considered economically sensitive and we worked hard to be able to release even these data.

Figure 8 now is labeled properly, moreover, we also labeled all figures as to whether they are single or multiple beam sonar data on the figure for clarity.

Reviewer 2: Calibration is described as integration of all values in a depth window for both MBES and SBES.

bubble-bubble acoustical interaction in this paper is due to scattering, not as in Tang et al., from acoustic coupling due to the compressibility of the bubbles. In the latter case, this is only for very high bubble density, which is not important, agreed. In the former case scattering can occur over any distance.

Respectfully, I disagree that a detailed quantitative theoretical consideration is needed. Ghosting is a very common artifact in sonar data where the true plume ghosted as a nearby faint plume. If multiscattering can create ghosts outside the plume, it seems evident that it also occurs inside. More to the point, and we apologize if the structure of the paper confused the discussion, Figure 9 shows that sonar return does not linearly scale with volume - there are non-linear effects, which are discussed at length. We thus argue to other researchers to not consider sonar data at only one depth, but to consider sonar data at multiple depths. Initially, we started to try and in fact do a more detailed theoretical investigation, but after writing out the equations, we realized that the number of unknowns is so large as to make the study only relevant for a mathematical paper not the real world. This is part of why we have removed equation 4, which was a remnant of the earlier effort.

With respect to resonance, if one looks at typical minor bubble plume size distributions - none of these plumes were of a magnitude to be major, bubble size distributions fall off extremely rapidly, an order of magnitude in 20-30% size change, e.g., see Fig. 10. These are not broad size distribution plumes. Thus, unless resonance just happens to be at the peak, most of the signal is going to come from non-resonance bubbles. Moreover, as shown in Fig. 10, the size rapidly changes as the bubble rises, so if all the response was from resonance, why is there no such trend in the sonar return with height in either the seep bubbles or in the engineered plume bubbles in the data presented herein (Fig. 7)?

Fig. 6, has been relabeled, as sigma is not in db, and the text now explains that the figure is included because of its trend, not its absolute value. The caption clearly says integrated – other figures are average.

WRT applying the results of the numerical model, the explanation is not more clear, and is in its own paragraph on this volume correction factor.

Regarding the title, see note above about permission for release.

Technically, a histogram is a probability distribution (or density) function. We now have added this terminology to occurrence to be more correct.

We have expanded our review to include other references, including Veloso et al. Unfortunately, Muyakshin and Sauter is beyond a paywall, and since I (and also co-authors) have left our universities, getting articles behind paywalls is quite a challenge – and this paper is not available on researchgate. I am not comfortable citing an article unless I have read it.

---

## Author Response (AR2)

**Editor Decision: Publish subject to minor revisions (Editor review)** (04 Feb 2017) by Dr. Nina Kirchner
Comments to the Author:
Your response to the reviewer's comments has now been evaluated.

While many comments have been addressed in the revised manuscript, some appear to have not been addressed:

* To shorten the introduction so that it becomes a concise introduction to the manuscript (suggested by Rev #1). Despite the suggestion to shorten, the revised manuscript is longer now by 2 pages!

• **We have removed 1200 words (shortening the introductory section by 1000 words), moved an additional figure to the supplemental material, and moved the bubble equations into the methodology where they describe the bubble model. The manuscript is now five pages shorter than the revised version.**

* To better justify the pre-study in Californian waters or consider removing it (Rev #1)
• **Study motivation rewritten to better explain motivation. Basically, the in situ data in the ESAS were less complete than desired and by including the Coal Oil Point data we are able to better explain the sonar observation of plume evolution.**

* To modify the title of the manuscript so that it better reflects its contents with regard to the geographical area covered (see Rev #2).
• **We have modified the title to: Sonar Gas Flux Estimation by Bubble Insonification: Application to Methane Bubble Flux from Seep Areas in the outer Laptev Sea**

Once these minor modifications are made, the paper can be published.

Non-public comments to the Author:
Dear Authors,

I have now evaluated your response to the reviewer's comments.
While many have been addressed in the revised manuscript, some appear to

have not been addressed.

I would like to suggest that you

* shorten the introduction so that it becomes a concise introduction to the manuscript (suggested by Rev #1). Despite the suggestion to shorten, the revised manuscript is longer now by 2 pages!

**We have removed 1200 words (shortening the intro section by 1000 words), moved an additional figure to the supplemental material, and moved the bubble equations into the methodology where they describe the bubble model. The manuscript is now five pages shorter than the revised version.**

* better justify the pre-study in Californian waters or consider removing it (Rev #1)
**Study motivation rewritten to better explain motivation. Basically, the in situ data in the ESAS were less complete than desired and by including the Coal Oil Point data we are able to better explain the sonar observation of plume evolution**

* modify the title of the manuscript so that it better reflects its contents with regard to the geographical areas covered (see Rev #2).
**• We have modified the title to: Sonar Gas Flux Estimation by Bubble Insonification: Application to Methane Bubble Flux from Seep Areas in the outer Laptev Sea**

* consider Muyakshin & Sauter, 2010 (Rev#2). I will send the pdf to you. You could have asked me for help in this issue!
**Thanks, I read it quickly. Muyakshin neglects that bubbles may multiply scatter – whereas Weber 2008 actually does the calculation and shows the multiple scattering is important. There are also a lot of other assumptions. For example, while it is true that sonar return decreases for bubbles smaller than resonance, this does not mean that they contribute nothing to volume flux and can be neglected. Muyakshin also did not have a measured bubble size distribution, so used others – but they were major bubble plumes, not minor bubble plumes. In a major bubble plume almost all the volume is carried by the very largest bubbles, which does not apply for the flows in our study. I could go on, but…..**

**However, I cite Muyakshin and Sauter as an example of using a ROV for a sonar survey, and also added a citation to Eberhardt's 2006 bubble measurements.**

Please make an effort to carry out these corrections so that the manuscript can be published afterwards.

Best wishes
Nina